# Preserving Plasticity in Continual Learning via Dynamical Isometry

Andries Rosseau [1]   Robert Müller [2]   Ann Nowé [1]

## Abstract

Continual training of deep neural networks under non-stationarity often leads to a progressive loss of plasticity, eventually limiting further learning. We relate plasticity to the empirical Neural Tangent Kernel, and identify dynamical isometry (the condition that layer-wise Jacobian singular values remain close to one) as a key mechanism for preserving plasticity in continual learning. We revisit a class of networks that are almost-everywhere isometric while remaining universal Lipschitz function approximators, demonstrating that near-dynamical isometry is compatible with expressive nonlinear representations. For general architectures, we propose an efficient isometry-promoting regularization scheme and identify a novel mechanism by which it can reactivate dormant ReLU units. Building on this, we introduce *AdamO*, an Adam-style adaptive optimizer that decouples isometry regularization from gradient updates, analogous to AdamW. We further reinterpret prior plasticity-preserving approaches through the lens of dynamical isometry, showing that they target only a partial measure of isometry. Across supervised and reinforcement-learning continual-learning benchmarks designed to induce plasticity loss, our methods consistently match or outperform existing approaches.

## 1. Introduction

Training a neural network typically optimizes parameters by gradient-based methods to minimize an expected loss. The effectiveness of this process depends strongly on the initialization of the network parameters, and a good initialization ideally places the network in a region of parameter space from which gradient descent can make rapid progress across a wide range of tasks (Martens et al., 2021; Saxe et al., 2013; Pennington et al., 2017; 2018; Xiao et al., 2018). For a single task, optimization often has ample time to move parameters to a region with low task loss. However, in the non-stationary setting of continual learning, inputs and losses may vary arbitrarily over time. In this setting, the learned parameters from one phase effectively become the initialization for the next, which might be a suboptimal starting point. Empirically, standard architectures and optimizers that succeed in single-task training often struggle under sustained non-stationarity, shown as a degradation in learning performance on new tasks (Dohare et al., 2024; Nikishin et al., 2022; Lyle et al., 2022; 2023; 2024b; Abbas et al., 2023). This gradual diminishing of a network's adaptive capabilities is commonly termed *plasticity loss*, and poses a major challenge for the development of continually learning systems.[1]

Several mechanisms have been implicated. In ReLU networks, units may become inactive ("dying ReLUs") and receive zero gradients (Sokar et al., 2023; Lu et al., 2019), or become perpetually active, reducing effective nonlinearity (Lyle et al., 2024b). Both correspond to pre-activations drifting to one side of the activation threshold. In reinforcement learning, *primacy bias* can arise when early bootstrapping targets dominate representation learning, limiting later adaptation (Nikishin et al., 2022). The optimization dynamics themselves can also cause plasticity loss. For example, the tendency of parameters to increase during learning can lead to overly sharp loss directions (high Hessian spectral norm), potentially making stable optimization difficult[2] (Lyle et al., 2023). Moreover, in scale-invariant networks, growing weight norms decrease the effective learning rate and make it increasingly difficult to change the direction of the weight vectors (Van Laarhoven, 2017; Lyle et al., 2024a).

While these failure modes are well documented, they are often studied in isolation and tied to specific architectures, losses, or tasks. In this paper, we argue that the common denominator is a deterioration in how efficiently gradient descent can update the network in all relevant output-space directions, expressed as an increase in anisotropy of the

---

[1]AI Lab, Vrije Universiteit Brussel [2]Aganthos. Correspondence to: Andries Rosseau <andries.rosseau@vub.be>.

*Proceedings of the 43rd International Conference on Machine Learning*, Seoul, South Korea. PMLR 306, 2026. Copyright 2026 by the author(s).

[1]Plasticity loss is related to, but different from, catastrophic forgetting, which concerns retention of past knowledge. We focus on the ability to acquire new knowledge.

[2]There is evidence indicating that gradient descent itself can limit this effect, though not remove it (Cohen et al., 2021).

empirical Neural Tangent Kernel (NTK) (Jacot et al., 2018). This leads to a functional task-agnostic definition of plasticity in terms of the local geometry of the parameter-function map. A finite network cannot have an isotropic empirical NTK for all input distributions simultaneously. However, under a task/input-agnostic prior where inputs are modeled as isotropic high-dimensional samples, a *dynamically isometric* network has an empirical NTK that concentrates near isotropy in expectation. Therefore, keeping the network near dynamical isometry yields a principled target for plasticity. A network is dynamically isometric when perturbations are propagated through its layers without systematic expansion or contraction, so that the singular values of the input-output Jacobian remain concentrated near one. This geometric condition has also been central to the signal propagation literature and is empirically known to lead to good network initializations (Saxe et al., 2013; Pennington et al., 2017; 2018; Xiao et al., 2018). Here, we connect it to plasticity and use it to derive plasticity-preserving architectures and training methods with the goal of driving the singular values of the layer-wise Jacobians to one. This helps gradient descent retain the capacity to make nontrivial progress from the current parameters, regardless of the current objective.

Concretely, we (i) define plasticity functionally and show that approximate dynamical isometry is a principled and tractable task-agnostic surrogate for NTK isotropy in finite networks; (ii) revisit almost-everywhere isometric architectures to establish that layer-wise isometry is compatible with universal Lipschitz approximation; (iii) propose an efficient isometry-promoting regularizer for general architectures, analyze a mechanism by which it revives dead ReLU units, and introduce *AdamO*, which decouples this regularization from the adaptive update in analogy to AdamW; and (iv) evaluate across supervised and RL continual-learning benchmarks, where promoting isometry preserves plasticity and yields strong continued performance.

## 2. Plasticity, the NTK and dynamical isometry

### 2.1. Defining plasticity

We define plasticity as the ability of a learning system to efficiently adapt its represented function toward minimizing achievable loss with respect to a distribution of tasks. Our definition applies to both the standard stationary single-task setting, as well as to non-stationary or multi-task settings.

Let $f_\theta : \mathcal{X} \to \mathbb{R}^{d_{\text{out}}}$ denote the learner's function represented by a model with parameters $\theta \in \mathbb{R}^P$, and let $\tau \sim \mathcal{T}$ denote a task sampled from a task distribution. A task $\tau$ specifies a data distribution $P_\tau(x, y)$ and a loss function $\ell_\tau(f(x), y)$, where $y = f_\tau^\star(x) + \epsilon$ with $f_\tau^\star$ the underlying target function and $\epsilon$ stochastic noise. The corresponding

population loss for task $\tau$ is

$$L_\tau(f) = \mathbb{E}_{(x,y) \sim P_\tau} \left[ \ell_\tau(f(x), y) \right]. \tag{1}$$

In the deterministic realizable setting ($\epsilon = 0$), minimizing the population loss to zero corresponds to recovering the unique target function $f_\tau^\star$ and therefore achieving perfect task generalization.

**Plasticity vs expressivity** Let $\mathcal{F} = \{f_\theta : \theta \in \mathbb{R}^P\}$ denote the induced function class. Since the target function may not be representable within $\mathcal{F}$, the optimal achievable population loss for task $\tau$ is

$$L_{\mathcal{F},\tau}^\star = \inf_{f \in \mathcal{F}} L_\tau(f),$$

and the corresponding optimal-loss set is

$$\mathcal{M}_{\mathcal{F},\tau} = \left\{ f \in \mathcal{F} : L_\tau(f) = L_{\mathcal{F},\tau}^\star \right\},$$

which need not contain a unique function. To separate plasticity from expressivity, we define plasticity in terms of the loss reduction with respect to the optimal *achievable* population loss, rather than convergence to the ideal target function. This distinction can be made explicit by noting that the loss difference for a task $\tau$ relative to the target function decomposes as

$$L_\tau(f_\theta) - L_\tau(f_\tau^\star) = \underbrace{L_\tau(f_\theta) - L_{\mathcal{F},\tau}^\star}_{\text{plasticity gap}} + \underbrace{L_{\mathcal{F},\tau}^\star - L_\tau(f_\tau^\star)}_{\text{approximation gap}}.$$

The approximation gap depends on the expressivity of the function class, whereas the plasticity gap measures the extent to which the learning dynamics fail to efficiently approach the optimal-loss set, i.e., reduce excess achievable loss.

**Resource efficiency** Intuitively, plasticity is not only about minimizing loss, but also about how efficiently this can be attained. Different learning systems may eventually reach similar performance while requiring different amounts of compute, data, optimization steps, memory, or energy. Plasticity should therefore contain some notion of resource-sensitivity. Exhaustive parameter search, for example, may eventually discover near-optimal solutions, but can require a prohibitively large amount of compute. To account for this, we characterize plasticity with respect to some update mechanism. Let

$$\theta_t^\tau = \mathcal{U}_{t,\tau}(\theta)$$

denote the parameter state obtained after adapting the initial state $\theta$ to task $\tau$ using $t$ resource units. Here $t$ may denote optimization steps, samples, wall-clock time, floating-point operations, energy, or another resource. In this work we focus mainly on first-order gradient-based optimization, where $t$ denotes the number of update steps.

**Definition 2.1** (Plasticity). Define the excess achievable loss, or plasticity gap, at parameter state $\theta$ by $\Delta_\tau(\theta) = L_\tau(f_\theta) - L^\star_{\mathcal{F},\tau}$. For a task $\tau$ with $\Delta_\tau(\theta) > 0$, define the fraction of eliminated excess achievable loss after budget $t$ as

$$\rho_{\tau,t}(\theta) = 1 - \frac{\Delta_\tau(\theta_t^\tau)}{\Delta_\tau(\theta)}.$$

The *plasticity of state* $\theta$ with respect to task distribution $\mathcal{T}$, update mechanism $\mathcal{U}$, and budget $t$ is then

$$\mathcal{P}_{\mathcal{T},t}(\theta) = \mathbb{E}_{\tau \sim \mathcal{T}}\left[\rho_{\tau,t}(\theta)\right]. \tag{2}$$

Thus, $\mathcal{P}_{\mathcal{T},t}(\theta)$ measures the expected fraction of excess achievable loss that can be eliminated using budget $t$. [3]

## 2.2. Task-agnostic plasticity and the NTK

We now relate plasticity under first-order gradient descent to the geometry of the empirical Neural Tangent Kernel (NTK) (Jacot et al., 2018). For a task dataset $D_\tau = \{(x_i, y_i)\}_{i=1}^{n_\tau}$, write $X_\tau = (x_1, \ldots, x_{n_\tau})$, and collect the network outputs into a vector

$$f_\theta(X_\tau) \in \mathbb{R}^m, \qquad m = n_\tau d_{\text{out}}.$$

Let

$$J_\theta(X_\tau) = \frac{\partial f_\theta(X_\tau)}{\partial \theta} \in \mathbb{R}^{m \times P}$$

be the parameter-output Jacobian, and let

$$g_\tau = \nabla_f L_\tau(f_\theta(X_\tau))$$

be the empirical loss gradient in output space. A gradient descent step in parameter space induces the first-order functional update

$$\Delta f = -\eta J_\theta(X_\tau) J_\theta(X_\tau)^\top g_\tau = -\eta K_\theta(X_\tau) g_\tau,$$

where

$$K_\theta(X_\tau) = J_\theta(X_\tau) J_\theta(X_\tau)^\top$$

is the empirical NTK. Hence, to first order,

$$\begin{aligned} \Delta L_\tau &= L_\tau(f_\theta + \Delta f) - L_\tau(f_\theta) \\ &= -\eta\, g_\tau^\top K_\theta(X_\tau) g_\tau + O(\eta^2). \end{aligned} \tag{3}$$

The quantity controlling local progress is therefore the Rayleigh coefficient

$$\mathcal{R}_{K_\theta(X_\tau)}(g_\tau) = \frac{g_\tau^\top K_\theta(X_\tau) g_\tau}{\|g_\tau\|_2^2}. \tag{4}$$

Large Rayleigh coefficients correspond to directions in which gradient descent can rapidly change the represented function. Small coefficients correspond to directions in which the network is locally rigid. Directions in the nullspace of $K_\theta(X_\tau)$ cannot be updated at all.[4]

**Task-agnostic plasticity** A natural notion of plasticity is *task-agnostic* plasticity, where we assume no prior over the task distribution. This means that any direction $g \in \mathbb{R}^m$ should be considered a possible task-gradient. Therefore, task-agnostic plasticity ideally requires first-order gradient descent to make comparable progress for all possible output-space directions, with no direction preferred in advance. The ideal task-agnostic NTK is consequently

$$K_\theta(X) \approx cI_m, \qquad c > 0 \tag{5}$$

for any $X$, meaning that the empirical NTK is isotropic for any set of inputs. Equivalently,

$$\mathcal{R}_{K_\theta(X)}(g) \approx c \qquad \text{for all } g \neq 0.$$

The overall scale of this progress cannot simply be made arbitrarily large, as this would lead to unstable dynamics. Standard stability conditions for gradient descent on an $L$-smooth objective require $\eta \lesssim 1/L$, so increasing $c$ forces $\eta$ to shrink proportionally, which cancels the amplification in the loss update. The optimal scaling can in principle be found using the local curvature/Hessian, but requires costly second order calculations. In practice, for first-order gradient descent, the learning rate is simply tuned for global trade-off between speed and stability.

More precisely, fix a stability budget $\eta \leq 1/\lambda_{\max}(K_\theta(X))$. The worst-case first-order progress over unit-norm task-gradient directions is then $\eta\, \lambda_{\min}(K_\theta(X))$, which is maximized exactly when $\lambda_{\min} = \lambda_{\max}$, i.e. when $K_\theta(X)$ is isotropic. Under first-order gradient descent, task-agnostic plasticity is therefore controlled by the *anisotropy* (condition number) of the empirical NTK rather than its scale: anisotropy forces the learning rate to respect the largest eigenvalue while progress along the smallest-eigenvalue directions stalls.

**Isotropy as a prior, not a frozen target** Task-agnostic isotropy is the idealized target only *before* a task is known, or equivalently as a property of the parameter state at the moment the new task is presented. Once a task arrives, gradient descent should and does specialize the NTK toward the task-relevant subspace; this evolution is precisely feature learning. We therefore will not aim to make $K_\theta$ strictly isotropic in practice. Rather, we keep the network near a well-conditioned NTK spectrum that balances pure plasticity and performance to both avoid that an output-space direction has already collapsed when the next, unknown task is presented, while also not prohibiting all feature learning.

---

[3] One can avoid choosing $t$ by integrating $P_{\mathcal{T},t}$ into a discounted plasticity $\mathcal{P}_{\mathcal{T}}^\gamma(\theta) = (1-\gamma)\sum_{t=0}^\infty \gamma^t \mathcal{P}_{\mathcal{T},t}(\theta)$, with $\gamma \in [0,1)$.

[4] Although Eqs. (3)–(4) are written for full-batch gradient descent, they hold for SGD in expectation: the minibatch gradient is unbiased, so taking the expectation over minibatch sampling gives $\mathbb{E}[\Delta f] = -\eta K_\theta(X)g$ and $\mathbb{E}[\Delta L_\tau] = -\eta\, g^\top K_\theta(X)g + O(\eta^2)$. The empirical NTK and its anisotropy therefore govern first-order progress for both GD and SGD.

## 2.3. Dynamical isometry

Exact isotropy of the empirical NTK for all input distributions is an idealization, and cannot hold for a finite network. Since

$$\text{rank}(K_\theta(X)) \le P,$$

any dataset with $m > P$ necessarily produces an NTK with a nontrivial nullspace. Even when $m \le P$, requiring $K_\theta(X) = cI_m$ for every possible $X$ would require the parameter-gradient features $\nabla_\theta f_{\theta,a}(x)$ to be mutually orthogonal for arbitrarily many distinct input-output pairs. A finite parameter space cannot support this. Thus exact input-agnostic NTK isotropy is unattainable.

We therefore relax the requirement from isotropy for every input dataset to isotropy in expectation over a task-agnostic input prior. Since a task-agnostic prior gives no preferred input direction, we take the input distribution to be isotropic in expectation. For normalized samples $x_i \in \mathbb{R}^d$, this gives

$$\mathbb{E}[XX^\top] = I_n, \qquad x_i^\top x_j = O_\mathbb{P}(d^{-1/2}) \quad (i \ne j),$$

so when $d$ is large, samples become nearly orthogonal. The goal under this relaxation then becomes

$$\mathbb{E}_X[K_\theta(X)] \propto I_m,$$

with finite-sample deviations controlled by the random correlations in $X$. We now ask what architectural condition preserves this property most directly.

Consider a depth-$L$ network

$$\begin{aligned} h_0(x) &= x, \\ z_l(x) &= W_l h_{l-1}(x) + b_l, \\ h_l(x) &= \phi_l(z_l(x)). \end{aligned} \tag{6}$$

Let $H_l \in \mathbb{R}^{n \times d_l}$ collect the activations $h_l(x_i)$ as rows. For output coordinate $a$, define

$$B_{l,a} \in \mathbb{R}^{n \times d_l}, \qquad (B_{l,a})_{i,:} = \frac{\partial f_{\theta,a}(x_i)}{\partial z_l(x_i)}.$$

For the weight parameters of layer $l$, the NTK block between output coordinates $a, b$ can then be written compactly as

$$K_l^{ab} = (H_{l-1} H_{l-1}^\top) \odot (B_{l,a} B_{l,b}^\top), \tag{7}$$

where $\odot$ denotes the Hadamard product. Summing over the layers gives the total NTK:

$$K_\theta^{ab}(X) = \sum_{l=1}^{L} K_l^{ab}, \tag{8}$$

with analogous bias terms depending only on the sensitivity Gram matrices.[5]

---

[5] The decomposition focuses on the weight contribution to the NTK. Bias parameters add terms of the form $B_{l,a} B_{l,b}^\top$. These obey the same backward-conditioning requirement and do not affect the main argument about preservation of input geometry through weight maps.

In order to keep the NTK isotropic under this isotropic input prior, the forward Gram matrices $H_l H_l^\top$ should preserve the (expected) isotropy of $XX^\top$ and the backward sensitivity Grams $B_{l,a} B_{l,b}^\top$ should not introduce output-directional anisotropy. This can be accomplished when the network has layer-wise input-output Jacobian singular values that are close to one, known as *dynamical isometry* (Saxe et al., 2013; Pennington et al., 2017; 2018; Xiao et al., 2018).

Layer-wise isometry is strictly stronger than end-to-end isometry of $J_L \cdots J_1$: isometry of each factor implies isometry of the product, but not conversely. The NTK needs the stronger version, since by (7)–(8) it is a sum of per-layer terms coupling the intermediate Gram $H_{l-1} H_{l-1}^\top$ with the backward sensitivity $B_{l,a} B_{l,b}^\top$, which stay well-conditioned only if every layer is isometric—an expanding layer cancelled by a contracting one leaves the intermediate Grams distorted even when the product is an isometry.

As an example, consider the deep linear case with square orthogonal weights and identity activations, where this condition holds exactly. We have

$$H_l H_l^\top = XX^\top \quad \text{for all } l,$$

and the layer-to-output maps are orthogonal, meaning

$$B_{l,a} B_{l,b}^\top = \delta_{ab} \mathbf{1} \mathbf{1}^\top.$$

Substituting into Eq. (7) yields

$$K_\theta^{ab}(X) \propto \delta_{ab} XX^\top,$$

or, equivalently,

$$K_\theta(X) \propto XX^\top \otimes I_{d_\text{out}}. \tag{9}$$

Thus, for an isometric network, the NTK inherits the input Gram geometry. Under the task-agnostic isotropic input prior,

$$\mathbb{E}_X[K_\theta(X)] \propto I_n \otimes I_{d_\text{out}},$$

and for high-dimensional sampled inputs the empirical NTK is close to this isotropic expectation up to the random off-diagonal correlations of $XX^\top$. Hence, layer-wise isometry is a principled and tractable surrogate to task-agnostic NTK isotropy, whereas no *sufficient* condition exists absent a prior over future tasks, owing to the unavoidable NTK nullspace. We note that our result complements insights from the theory of random network initialization (Pennington et al., 2017; 2018; Martens et al., 2021; Saxe et al., 2013; Xiao et al., 2018). This line of work analyzes semi-formally how depth affects signal and gradient propagation at initialization, and shows that training is facilitated when networks are initialized near dynamical isometry. This encourages singular values of the relevant input-output Jacobians to remain concentrated near one, so that signals and gradients neither

systematically expand nor contract across depth. We extend this viewpoint by treating dynamical isometry not only as an initialization target for *random* networks, but as a geometric condition whose drift provides insight into plasticity loss.

For nonlinear networks, exact data-independent dynamical isometry is generally impossible because the Jacobians depend on the activation pattern. The practical target is therefore *approximate* dynamical isometry: keep the layerwise Jacobian spectra close to 1 by controlling the weights and the activations. In the next sections, we will soften strict isometry to allow for more expressive networks, while still aiming to keep close to the isometric manifold in order to preserve plasticity.

# 3. Isometry and expressivity

A natural concern is whether staying close to an (approximately) isometric manifold is actually compatible with expressive nonlinear hypothesis classes. Pennington et al. (2018) show that for random networks at initialization, strict dynamical isometry is only possible when the network collapses to a linear (orthogonal) network. However, for practical networks, we do not require strict isometry. Here, we first address expressivity near the isometric manifold by briefly revisiting a class of networks that are isometric almost-everywhere, providing an *existence proof* of architectures that are (i) (almost-everywhere) isometric in the layer-wise sense relevant for task-agnostic plasticity, yet (ii) remain universal approximators over a rich function class.

## 3.1. An almost-everywhere isometric universal class.

Consider fully-connected networks of the form

$$
\begin{aligned}
h_0(x) &= x, & x \in \mathbb{R}^{d_{in}} \\
h_\ell(x) &= \phi(W_\ell h_{\ell-1}(x) + b_\ell), & \ell = 1, \ldots, L, \quad (10) \\
f_\theta(x) &= h_L(x),
\end{aligned}
$$

where each linear map is *orthonormal* (all singular values equal 1; in the square case $W_\ell$ is orthogonal) and the activation $\phi$ is *GroupSort*, as proposed in Anil et al. (2019); Chernodub & Nowicki (2016). GroupSort partitions preactivations into fixed-size groups and sorts within each group. For groups of size two, it is called *MaxMin*. These non-linear activations are 1-Lipschitz, and wherever they are differentiable their input-output Jacobian is a (block) permutation matrix. Consequently, at points of differentiability, $\nabla \phi(z)$ is orthogonal and hence norm-preserving.

Let $\mathbf{J}_\ell(x) = \nabla_{h_{\ell-1}} h_\ell(x)$ denote the layer-wise input–output Jacobian. For GroupSort networks with orthonormal $W_\ell$, at any $x$ where all layers are differentiable we have the factorization

$$
\mathbf{J}_\ell(x) = P_\ell(x) W_\ell, \tag{11}
$$

where $P_\ell(x)$ is a (block) permutation matrix induced by the local sorting pattern. Therefore,

$$
\mathbf{J}_\ell(x)^\top \mathbf{J}_\ell(x) = \mathbf{I} \tag{12}
$$

and hence the singular values $\sigma_i$ are

$$
\sigma_i(\mathbf{J}_\ell(x)) = 1 \ \forall i, \tag{13}
$$

i.e., each layer is an isometry at such $x$. The set of non-differentiability corresponds to within-group ties (changes in sorting order). This set is contained in a finite union of affine hyperplanes in pre-activation space and is therefore Lebesgue-null. Equivalently, the network is differentiable and satisfies (13) *almost everywhere* (a.e.) on $\mathbb{R}^{d_{in}}$. In particular, for any data distribution $\mathcal{D}$ absolutely continuous w.r.t. Lebesgue measure, these isometry statements hold $\mathcal{D}$-almost surely.

**Implication for loss-agnostic plasticity.** Recall the backpropagation (suffix) Jacobians $\mathbf{B}_\ell(x) = \nabla_{h_\ell} f_\theta(x) = \mathbf{J}_L(x) \cdots \mathbf{J}_{\ell+1}(x)$. If every $\mathbf{J}_k(x)$ is orthogonal (or, more generally, an isometry in the sense $\mathbf{J}_k(x)^\top \mathbf{J}_k(x) = \mathbf{I}$), then $\mathbf{B}_\ell(x)$ is also an isometry, and for every output direction $v \in \mathbb{R}^{d_{out}}$,

$$
\|\mathbf{B}_\ell(x)^\top v\|_2 = \|v\|_2 \qquad \text{for a.e. } x. \tag{14}
$$

Concretely, this means GroupSort networks (with orthogonal linear layers) preserve norms, but not necessarily angles. There are no vanishing or exploding gradient signals, but gradients for different samples can still become correlated or anti-correlated, which is necessary for feature learning.

## 3.2. Expressivity of isometric a.e. networks

The apparent simplicity of sorting-based activations is misleading. GroupSort networks are highly expressive, for reasons closely analogous to the expressivity of elementwise piecewise-linear activations, but without sacrificing norm preservation. For example, ReLU achieves expressivity through input-dependent gating: different activation patterns induce a combinatorial number of affine linear regions, at the cost of locally collapsing certain directions and reducing the rank of the Jacobian. GroupSort induces a similar combinatorial partition of input space, but replaces gating with input-dependent permutations (reflections in the case of MaxMin) of coordinates within fixed-size groups. The network is therefore still piecewise linear, with linear regions indexed by sorting patterns, but each region is associated with an orthogonal (norm-preserving) map rather than a rank-deficient projection. Expressivity arises from the rapid growth in the number of such regions with depth, while all directions are preserved locally.

As Anil et al. (2019) show, orthogonal linear maps composed with GroupSort activations are dense in the class of 1-

Lipschitz functions on compact domains (a universal approximation theorem based on a restricted Stone–Weierstrass argument). This establishes the existence of highly expressive solutions within architectures that are (a.e.) isometric at the layer level. Existence alone, however, does not guarantee that such solutions are readily reached by optimization. In our experiments, we show that gradient descent on almost-everywhere isometric networks reliably finds functions that are expressive enough to match, and in some settings surpass, the performance of unrestricted architectures.

# 4. Promoting isometry through regularization

The preceding sections motivate (approximate) layer-wise dynamical isometry as a proxy for preserving task-agnostic plasticity under sustained non-stationarity. We now turn to more general architectures with element-wise activations, for which exact isometry is generally unattainable (without linearization), and discuss a regularization scheme that keeps the linear components of each layer close to a partial isometry by driving their singular values toward one.

## 4.1. Isometry of weight matrices

Consider a standard feedforward layer of the form

$$h_\ell(x) = \phi(z_\ell(x)), \qquad z_\ell(x) = W_\ell h_{\ell-1}(x) + b_\ell,$$

with elementwise nonlinearity $\phi$ and weight matrix $W_\ell \in \mathbb{R}^{d_\ell \times d_{\ell-1}}$. At inputs where $\phi$ is differentiable, the layer Jacobian factorizes as

$$\begin{aligned} \mathbf{J}_\ell(x) &= \nabla_{h_{\ell-1}} h_\ell(x) = D_\ell(x) W_\ell, \\ D_\ell(x) &:= \mathrm{Diag}\big(\phi'(z_\ell(x))\big). \end{aligned} \tag{15}$$

Since $D_\ell(x)$ depends on both the data and the current parameters through $z_\ell(x)$, any attempt to enforce $\mathbf{J}_\ell(x)^\top \mathbf{J}_\ell(x) \approx I$ uniformly over $x$ is necessarily limited by the variability (and potential rank-deficiency) of $D_\ell(x)$. However, we can still control the multiplicative part $W_\ell$ and thereby prevent the weights from introducing additional ill-conditioning beyond that already imposed by the activation gates.

This statement can be made precise at the level of singular values and condition numbers. For any (fixed) diagonal $D$ and matrix $W$, submultiplicativity of the singular values yields $\sigma_{\max}(DW) \leq \sigma_{\max}(D)\,\sigma_{\max}(W)$, and $\sigma_{\min}(DW) \geq \sigma_{\min}(D)\,\sigma_{\min}(W)$, and hence (when $\sigma_{\min}(D) > 0$) the condition number, defined as $\kappa = \sigma_{max}/\sigma_{min}$, yields $\kappa(DW) \leq \kappa(D)\kappa(W)$. Thus, among all choices of $W$, the choice that minimizes distortion of gradient transport is precisely $\kappa(W) = 1$, i.e., $W$ is (pseudo-)orthogonal. In short: without architectural control over $D_\ell(x)$ (as we did for GroupSort networks), the best general-purpose target is to make $W_\ell$ isometric, so that the layer's conditioning is dominated by the nonlinearity rather than by weight drift.

## 4.2. Regularizing toward (pseudo-)orthogonality

We implement this principle via a differentiable penalty that drives the singular values of each $W_\ell$ toward one. A square matrix is orthogonal iff $W_\ell^\top W_\ell = W_\ell W_\ell^\top = I$. For rectangular matrices, only one of these constraints can hold.

Accordingly, for each layer we define the *Gram deviation* penalty

$$\mathcal{R}_{\mathrm{iso}}(W_\ell) := \begin{cases} \left\| W_\ell^\top W_\ell - I_{d_{\ell-1}} \right\|_F^2, & d_\ell \geq d_{\ell-1}, \\ \left\| W_\ell W_\ell^\top - I_{d_\ell} \right\|_F^2, & d_\ell < d_{\ell-1}. \end{cases} \tag{16}$$

This objective directly penalizes deviations of the *squared* singular values from one. For example, when $d_\ell \geq d_{\ell-1}$,

$$\left\| W_\ell^\top W_\ell - I \right\|_F^2 = \sum_{i=1}^{d_{\ell-1}} \left( \sigma_i(W_\ell)^2 - 1 \right)^2, \tag{17}$$

and analogously in the wide case with $W_\ell W_\ell^\top$. Hence minimizing (16) encourages $\sigma_i(W_\ell) \approx 1$ for all nonzero singular values, directly promoting approximate isometry and good conditioning. Moreover, the penalty (16) is efficient: it requires only matrix multiplications (no SVD), and its gradient has a closed form.

We note that this form of regularization is not novel to our work and has been successfully employed in single-task learning settings (Bansal et al., 2018; Li et al., 2019; Huang et al., 2020). Here, we extend its application to plasticity and continual learning. We refer to Appendix A for implementation details.

Putting things together, our training objective at time $t$ is

$$\min_\theta \ \mathbb{E}_{(x,y)\sim P_t}\big[\ell(f_\theta(x), y)\big] \ + \ \lambda \sum_{\ell=1}^{L} \mathcal{R}_{\mathrm{iso}}(W_\ell), \tag{18}$$

with $\lambda \geq 0$ controlling the strength of the isometry-promoting term. While (18) does not guarantee dynamical isometry in the presence of strongly saturating or gating nonlinearities, it prevents a prominent and empirically common failure mode in continual learning: progressive growth of weight norms and condition numbers that drives Jacobian singular values away from one and collapses gradient transport.

## 4.3. Revival of dead ReLUs

A key feature of (18) is that the orthogonality regularizer acts directly on each weight matrix, rather than propagating through downstream nonlinearities. This is relevant in ReLU networks, where dead units (strictly negative pre-activations on the current stream) receive zero task gradient. If we write $W_\ell = [w_1^\top; \ldots; w_{d_\ell}^\top]$, then even if the task gradient for some weight vectors $w_i$ is zero, the Gram-deviation penalty

generally still produces a nonzero update unless $w_i$ already has unit norm and is orthogonal to the other weight vectors. Consequently, as the subset of active neurons moves under the task loss, the regularizer couples the rows of $W_\ell$, rotating inactive $w_i$ to maintain an approximately orthonormal set.

Under non-stationarity, this coupling provides a plausible reactivation mechanism: as long as a subset of neurons remain active and move under the task loss, the orthogonality penalty tends to push the remaining weight vectors to be orthogonal to the active span. Unless the active subspace evolves entirely within the hyperplane orthogonal to all dead weights, this redistribution can move some previously dead units back into regions with positive pre-activations. The effect is indirect and does not guarantee revival in adversarial settings where $D_\ell(x)$ collapses on all inputs, but it supplies a mechanism absent from purely task-driven updates and matches our empirical finding that orthogonal regularization can sustain ReLU activity during continual learning (Section 6).

### 4.4. Orthogonal adaptive optimization: AdamO

Adaptive optimizers such as Adam (Kingma, 2014) are widely used in deep learning. A subtle practical issue is that if one naively optimizes the composite objective (18) with Adam, the moment estimates are built from the *sum* of task and regularizer gradients. Since the statistics and geometry of $\nabla \mathcal{R}_{\mathrm{iso}}$ differ substantially from those of the task gradient (it is dense, layer-coupled, and targets a specific matrix manifold), mixing them inside the adaptive preconditioner can undesirably rescale task updates or make the effective regularization strength highly parameter- and time-dependent.

We therefore propose *AdamO*, which decouples isometry updates from the adaptive gradient step, in direct analogy to decoupled weight decay (Loshchilov & Hutter, 2017). Concretely, let $g_t := \nabla_\theta \ell_t(\theta)$ be the stochastic task gradient and $r_t := \nabla_\theta \sum_\ell \mathcal{R}_{\mathrm{iso}}(W_\ell)$ the regularizer gradient. AdamO updates Adam's moments using $g_t$ only:

$$m_t = \beta_1 m_{t-1} + (1 - \beta_1) g_t,$$
$$v_t = \beta_2 v_{t-1} + (1 - \beta_2) g_t \odot g_t, \qquad (19)$$

and then applies a parameter update with a decoupled isometry step

$$\theta_{t+1} = \theta_t - \eta \frac{\hat{m}_t}{\sqrt{\hat{v}_t} + \varepsilon} - \eta_{\mathrm{iso}} \lambda \, r_t, \qquad (20)$$

where $\hat{m}_t, \hat{v}_t$ are the bias-corrected moments, $\eta$ is the base learning rate, and $\eta_{\mathrm{iso}}$ is an isometry step size (set equal to $\eta$ by default, but tunable).

Computing $\mathcal{R}_{\mathrm{iso}}(W_\ell)$ and its gradient requires only the Gram product $W_\ell^\top W_\ell$ (no SVD), at cost $O(d_\ell d_{\ell-1}^2)$ per

layer (comparable to a single forward pass when the batch size is of order $d_{\ell-1}$) plus one additional Gram matrix in memory. In our experiments this amounted to a few percent of wall-clock time and $\approx$4–5% memory.

## 5. Analysis of related work

A growing body of work addresses plasticity loss by identifying specific training pathologies (e.g., dead units, primacy bias, parameter growth) or by regularizing the network toward a "trainable" reference state like initialization. In this section, we review the key approaches used as baselines in our evaluation and demonstrate that many can be reinterpreted as targeting a restricted notion of dynamical isometry.

**Normalize-and-Project (NaP).** Lyle et al. (2024a) propose the combination of layer normalization (Ba et al., 2016) with a regular rescaling of the Frobenius norm of the weight matrices toward its value at initialization. The authors motivate this as stabilizing the effective learning rate in scale-invariant networks. Empirically, NaP is a competent plasticity-preserving baseline and outperforms many common techniques. Notably, when combined with earlier heuristics (ReDo (Sokar et al., 2023), regenerative regularization (Kumar et al., 2023), shrink-and-perturb (Ash & Adams, 2020)), the performance gap between methods becomes negligible, suggesting that NaP addresses a dominant failure mode and largely equalizes remaining differences. We relate this result to our work by noting that resetting the Frobenius norm of the weights to its initialization value to stabilize gradients is equivalent to keeping the *mean squared singular value* of each weight matrix constant (often near 1). In contrast, isometry (i.e., orthogonality) requires *all* squared singular values to be one. Thus, while NaP stabilizes the *average* gradient scale (in combination with the stabilizing and loss-smoothing effect of layer normalization (Ba et al., 2016)), isometry stabilizes *all* gradient directions[6]. Additionally, NaP offers its own ReLU revival pathway: since normalization combines signals from all pre-activations, it can route gradient signals from active neurons back to dead ones. Our revival mechanism (Section 4.3) is related but distinct: the movement of active neurons in the embedded space causes the other (dead) neurons to "follow", potentially reviving them. Finally, we note that NaP is tied to its architecture: its projection step relies on scale-invariance, whereas our isometry regularization applies broadly to linear/convolutional operators without requiring normalization.

**ReLU recycling.** ReDo (Sokar et al., 2023) recycles dormant/dead neurons in deep RL by resetting them to their

---

[6]A directly analogous distinction exists in the history of network initialization: classic Gaussian initialization (He et al., 2015) controlled only the *mean* signal variance, whereas orthogonal initialization controls all signal directions identically.

original weight when they are below an activity criterion. From our perspective, this can be viewed as restoring a subset of units to form a partial isometry (assuming standard orthogonal initialization). An obvious drawback is that resetting discards learned information; in environments where previously useful features reoccur at later times, such recycling may have exacerbated forgetting.

**L2 Init.** Regenerative regularization (Kumar et al., 2023), or L2 Init, adds an $\ell_2$ penalty toward the initial parameters. Again, when assuming orthogonal initialization, this promotes dynamical isometry. A potential limitation is reduced expressivity if good solutions lie far from the initialization basin; however, in sufficiently wide regimes, NTK-style arguments (Jacot et al., 2018) suggest that remaining close to initialization can still yield adequate function classes.

**Spectral norm regularization.** Closest to our work, Lewandowski et al. (2025) propose to regularize the maximal singular value of each weight matrix toward 1, motivated by the observation that the norms of the weight matrices grow during training. They acknowledge that preservation of the spectral properties of the weights at initialization can potentially be beneficial for continual learning and promote gradient diversity. However, they eventually only regularize the maximal singular value of the weight matrices (i.e., $\sigma_{max}(W_\ell) \to 1$), while we propose to directly control the *entire spectrum* of singular values (i.e., $\sum_{i=1}^{d_{\ell-1}} \left(\sigma_i(W_\ell)^2 - 1\right)^2 \to 0$, cf. (17)). Notably, the authors find empirically that squaring the spectral norm results in improved performance. This finding is explained by our framework: minimizing the squared spectral norm deviation $\sigma_{max}^2 - 1$ becomes mathematically equivalent to minimizing solely the largest term in the sum of (17).

**Weight decay.** We briefly mention weight decay/L2 regularization (Krogh & Hertz, 1991; Loshchilov & Hutter, 2017) as a classic regularization technique. From the perspective of controlling singular values, weight decay explicitly drives the mean squared singular value of weight matrices toward 0. When combined with layer normalization, it can perform adequately in continual learning (Lyle et al., 2024b), but it needs careful finetuning. This finding is again explained by our framework: weight decay pushes singular values to 0, and not to 1, so it can easily over- or undershoot the "correct" target. NaP directly sets this target to 1 and outperforms weight decay, so we refer to NaP for our baselines.[7]

## 6. Experiments

To demonstrate that remaining close to dynamical isometry reduces plasticity loss, we evaluate combinations of the following architectural components across both fully connected (MLP) and convolutional (CNN) networks:

---

[7]For experiments, see Figures 5, 6, and 4.

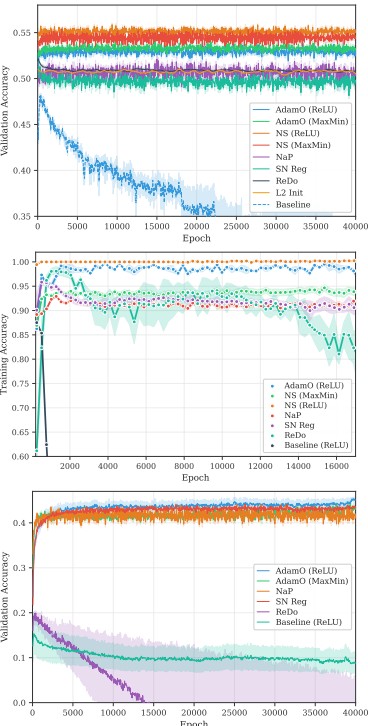

*Figure 1.* Top to bottom: Results for Pixel-Permutation (40 epochs/task, 1000 tasks, batch size 250), Random-Label Memorization (256 epochs/task, 200 tasks, batch size 128) and Label-Shuffling (40 epochs/task, 1000 tasks, batch size 250). Results for ReDo and L2 Init have similar noise profiles as e.g., NaP, but are smoothed for Pixel-Permutation to not clog the figure. Pixel-Permutation and Random-Label Memorization use MLPs, Label-Shuffling uses CNNs. We refer to Appendix B for architecture and implementation details.

(i) **Strictly orthogonal linear operators**, implemented via the differentiable Newton–Schulz method (Björck & Bowie, 1971; Huang et al., 2020; Grishina et al., 2025).

(ii) **Soft-orthogonal linear operators**, subject to regularization via our proposed AdamO optimizer (Section 4.4).

(iii) **GroupSort activations**, specifically MaxMin (Anil et al., 2019), which applies sorting on binary feature partitions and is isometric almost-everywhere.

(iv) **ReLU activations**, included to assess compatibility with (near-)orthogonal weights and to evaluate the proposed ReLU revival mechanism (Section 4.3).

We evaluate these methods in continual supervised and reinforcement learning settings. Overall, our methods consistently maintain plasticity and yield strong continued performance, consistently matching or surpassing prior methods. For practical use we recommend **AdamO + ReLU**: it adds little compute over standard Adam (Section 4.4), and ReLU's high expressive efficiency together with our revival mechanism (Section 4.3) provide strong performance.

Newton-Schulz is used as the orthogonal limit of AdamO.

## 6.1. Supervised learning experiments

We validate our approach on three benchmarks widely used to quantify plasticity loss. First, we use Random-Label Memorization (CIFAR-10) (Zhang et al., 2016) to test the network's raw capacity to fit arbitrary data streams; since generalization is impossible on random noise, we report training accuracy to strictly measure optimization plasticity. Second, we evaluate on Permuted MNIST/CIFAR-10 (Goodfellow et al., 2013) and Label-Shuffled CIFAR-100 (Ash & Adams, 2020), which test adaptation to distribution shifts. For these, we report performance on held-out validation sets, verifying that our method preserves generalization capability rather than just memorization. Since we use sufficiently expressive networks, the optimal achievable loss $L^\star_{\mathcal{F},\tau} \approx 0$, so the plasticity gap $\Delta_\tau(\theta) \approx L_\tau(f_\theta)$ and continued task performance directly tracks the plasticity metric $\mathcal{P}_{\mathcal{T},t}$ of Eq. (2). As shown in Figure 1, our methods consistently outperform competing approaches across all three scenarios, with narrower margins on Label-Shuffled due to performance saturation. This confirms that promoting dynamical isometry not only maintains the raw plasticity required to fit new data (Random Labels) but also supports generalization to unseen data (Permuted/Shuffled). We additionally conducted a preliminary small-scale transformer study on the continual CIFAR-10 pixel-permutation benchmark, with diagnostics and discussion provided in Appendix B.7.

In Appendix B.4, Figure 7, we show that our methods revive dead ReLU units, substantiating our revival mechanism outlined in Section 4.3. We also include metrics such as condition number, effective NTK rank, Jacobians, and weight vectors in Appendix B.3 (Figures 5–6). Our methods improve conditioning and rank diagnostics across these categories, supporting our theoretical arguments on plasticity.

## 6.2. Reinforcement Learning experiments

We consider continual reinforcement learning (RL), where non-stationarity arises both from inherent policy shifts during training and from external changes to the environment. We use Proximal Policy Optimization (PPO) (Schulman et al., 2017) as the base algorithm for all experiments.

First, we use a modified, continual version of MinAtar (Young & Tian, 2019) to evaluate MLP architectures. The agent cycles through miniaturized versions of four Atari games for 20 cycles (15M steps/game), with the next game sampled randomly at each switch (balanced to ensure uniform coverage). To further heighten non-stationarity, we randomly permute the observation channels at the onset of each new game instance. Results in Figure 2 show that while the PPO baseline loses its ability to learn over time, AdamO, combined with either ReLU or MaxMin (GroupSort) acti-

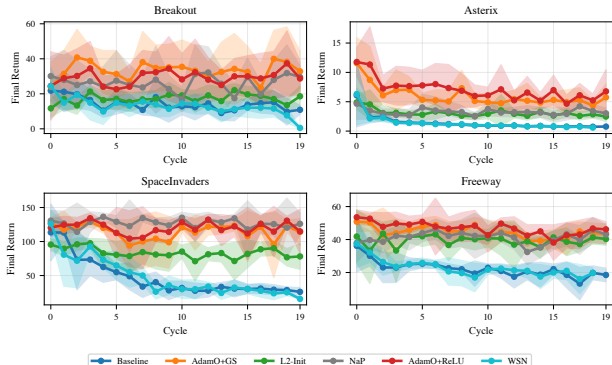

*Figure 2.* Continual MinAtar with random channel permutations. Results are reported over 8 random seeds. Each environment runs for 15 million steps per cycle for a total of 1.2B steps. GS is GroupSort/MaxMin.

vations, maintains consistent performance throughout the 20 cycles. Our method matches or outperforms the baselines on regular evaluation runs. (We detail explicit learning curves and additional RL diagnostics in Appendix B; see Figures 9–15, 18–23.)

Next, we consider convolutional architectures (CNNs) using a continual version of Octax (Radji et al., 2025), which emulates classic arcade games with dynamics similar to the Arcade Learning Environment (Bellemare et al., 2013). Here, the agent cycles through eight distinct games for three full repetitions in a fixed order. The results for continual Octax (Figure 17) reaffirm our findings: AdamO combined with GroupSort or ReLU consistently preserves plasticity and achieves superior cumulative rewards compared to standard approaches. Additional Octax diagnostics are provided in Appendix B.6, Figures 18–23.

## 7. Discussion

We characterized plasticity functionally, tied it to the geometry of the empirical NTK, and identified approximate dynamical isometry as a tractable surrogate under task uncertainty—compatible with expressive nonlinear function classes via almost-everywhere isometric networks and orthogonality-promoting regularization. A promising future direction is genuinely deep reinforcement learning– networks are typically kept shallow because depth tends to destabilize training. Since dynamical isometry was originally introduced to enable signal propagation through very deep networks, promoting it may help unlock stable training of deeper RL agents. A second is large language models, where isometry could improve pre- and post-training: skip connections keep models near isometry at initialization, but residuals erode it as their contributions grow over time. Isometry-preserving methods may improve depth-related issues, or improve continual learning.

## Impact Statement

This paper presents work whose goal is to advance the field of Machine Learning. There are many potential societal consequences of our work, none which we feel must be specifically highlighted here.

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

# Appendix

## A. Implementation details for regularizing toward (pseudo-)orthogonality

A square matrix is orthogonal iff $W_\ell^\top W_\ell = W_\ell W_\ell^\top = I$. For rectangular matrices, only one of these constraints can hold. Satisfying $W_\ell^\top W_\ell = I$ (when $d_\ell \geq d_{\ell-1}$) makes $W_\ell$ an isometric embedding, while satisfying $W_\ell W_\ell^\top = I$ (when $d_\ell \leq d_{\ell-1}$) makes it an isometry on the output space (equivalently, $W_\ell^\top$ is an isometric embedding), which is particularly relevant for backward signal transport.

For convolutional layers, we follow the standard convention of reshaping the kernel tensor into a 2D matrix (e.g., $(k^2 c_{\text{in}}) \times c_{\text{out}}$) and applying the same penalty to encourage diversity/orthogonality across filters. Biases do not enter the layer Jacobian and are therefore not directly constrained by isometry; in our experiments we either leave them unregularized or apply a weak standard penalty for numerical stability.

## B. Extra experiments

### B.1. Supervised learning: architectural details

The architectures used for learning are outlined below (finetuned with hyperparameter sweeps). We use orthogonal initialization of weights in all architectures. When not explicitly specified otherwise, we use ReLU activations.

**Multi-layer Perceptron:** We use standard fully connected feed forward networks with depth 4 and size 512 for each hidden layer. Learning rate for Adam and AdamO is 1e-4. We run 8 seeds per method.

**Convolutional Neural Networks:** We use a combination of 3 convolutional layers, with each 32 channels (3 x 3), with strides [1, 1, 2], followed by 2 fully connected layers, with hidden size 256. Learning rate for Adam and AdamO is 1e-3. We run 8 seeds per method.

For Lipschitz-1 networks, we leave the output head unrestricted (or regularize it with a softer regularizer) so the network can approximate L-lipschitz functions.

**Algorithm hyperparameters:**

- For AdamO, we use a regularization strength of 1e-3 for the orthogonal penalty. Learning rate is kept the same as for the task loss.

- Newton-Schulz orthogonalization is run through the network's forward pass for 7 iterations per weight matrix, with spectral norm guardrailing using 1 power iteration. See (Anil et al., 2019). Every 10 000 (arbitrarily set, any reasonable number works) training steps, the "raw" weights are orthogonalized using Newton-Schulz, to ensure gradients remain well-conditioned on the raw weights too.

- For ReDo, we use $\tau = 0.1$, and apply ReDo every 1000 steps with a batch size of 64. These are the standard settings, and worked best in our experiments.

- For NaP, we use LayerNorm with fixed scale (1 or $\sqrt{2}$) and offset (0), which performed better in our experiments compared to regularizing them (Lyle et al., 2024a). We project either every 1000 or 10 000 steps (we observed no functional difference).

- L2 init: regularization strength: 1e-3.

- Spectral norm regularization strength: 1e-3, and we used one power iteration to calculate largest singular value of the weight matrices.

### B.2. Sensitivity to the orthogonal regularization strength

We include an ablation over the orthogonal-regularization strength $\lambda$ for AdamO on the continual CIFAR-10 random-label benchmark. As expected, the regularization strength should be tuned for best performance, since too little regularization

does not sufficiently preserve conditioning, while too much regularization can begin to interfere with task optimization. At the same time, the ablation indicates a reasonably broad good operating regime rather than a single sharp optimum, suggesting that AdamO is practically robust once $\lambda$ is chosen in the appropriate range.

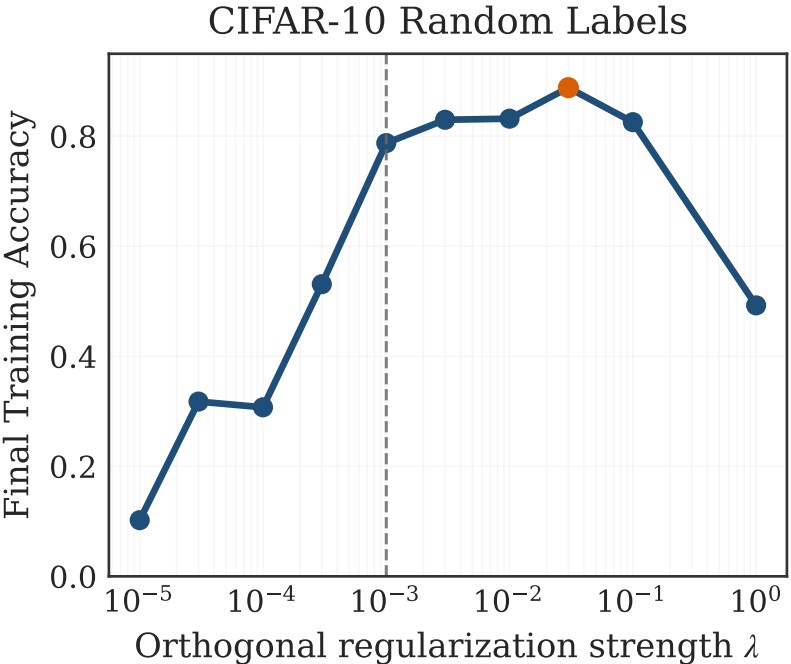

*Figure 3.* Sensitivity of AdamO to the orthogonal-regularization strength $\lambda$ on the continual CIFAR-10 random-label benchmark.

### B.3. Additional diagnostic metrics

We provide additional diagnostics for the CIFAR-10 pixel-permutation benchmark in Figures 4, 5, and 6, focusing respectively on the weight spectra, empirical NTK statistics, and intermediate Jacobian spectra. These figures complement the performance plots in the main text by showing that the methods which preserve plasticity also maintain better-conditioned dynamics and richer effective dimensionality throughout training.

Across these supervised diagnostics, our methods are favorable essentially across the board. In Figure 4, they maintain tighter weight spectra and higher effective rank; in Figure 5, the empirical NTK remains better conditioned and closer to isotropic; and in Figure 6, the Jacobian spectra stay more stable throughout training. Taken together, these trends support the main claim of the paper: the methods that preserve plasticity are also the ones that better preserve dynamical isometry and avoid progressive spectral collapse.

### B.4. Dead ReLU revival

To make the dormant-unit mechanism explicit, we include the dedicated dormant-unit diagnostic in Figure 7. The figure tracks how the number of Sokar-style dormant ReLU units evolves during training, and highlights that the isometry-preserving methods substantially reduce or reverse the buildup of inactive features relative to the baselines. This provides direct empirical support for the revival mechanism discussed in Section 4.3.

### B.5. Minatar

The RL diagnostics mirror the supervised picture. For both MinAtar and Octax, the methods that perform best also tend to keep the spectra, Jacobian statistics, and NTK metrics in a visibly healthier regime over time. In particular, they generally preserve broader effective rank, better conditioning, and lower dormant-neuron buildup, which is consistent with the

*Table 1.* Experimental setup for continual MinAtar.

| Network | |
|---|---|
| Architecture | MLP: 64-64-64-64 |
| Activation | ReLU |
| Initialization | Orthogonal ($\sqrt{2}$), output: 0.01 |
| **PPO** | |
| Parallel environments | 2048 |
| Rollout length | 128 |
| PPO epochs | 4 |
| Minibatches | 128 |
| Learning rate | $2.5 \times 10^{-4}$ (constant) |
| GAE $\lambda$ | 0.95 |
| Clip $\epsilon$ | 0.2 |
| Entropy coefficient | 0.01 |
| Value coefficient | 0.5 |
| **Continual Learning** | |
| Wrapped Observation Shape | $(10, 10, 10)$ |
| Wrapped Action Shape | 6 |
| Steps per game | 15M |
| Training cycles | 20 |
| Seeds | 8 |
| Games | Breakout, Asterix, SpaceInvaders, Freeway |

interpretation that plasticity loss in RL is likewise accompanied by a gradual deterioration of the network's local geometry.

We provide full learning curves for all configurations over all cycles in figure 16.

The experiments hyperparameters are provided in table 1.

### B.6. Octax

We consider the environemnts "brix", "submarine", "filter", "tank", "blinky", "missile", "ufo", "wipe_off" in this sequence for 3 cycles with 5 million training steps per env. We utilise PPO with shared backbone following the implementation provided in (Radji et al., 2025). A precise list of our hyperparameters is given in table 2.

### B.7. Attention and transformers

We also include a preliminary transformer study on the continual CIFAR-10 pixel-permutation benchmark. The setup is as follows: a 4-block transformer with 4 attention heads per block and MLP sublayers, evaluated across vanilla training, LayerNorm, LayerNorm with weight decay, and AdamO, all with skip connections. These experiments are intentionally small-scale and should be interpreted as an initial diagnostic rather than a final statement on large transformer training. In particular, attention is typically contractive and residual-path interactions with normalization and weight decay have become increasingly important in modern deep transformers. A larger-scale study on deeper architectures and LLM-style fine-tuning remains an important direction for future work.

Even with those caveats, the results are encouraging. The validation-accuracy curves show that AdamO remains stable and competitive at learning rates where vanilla baselines degrade and where LayerNorm and weight decay alone do not provide the same degree of robustness. The input-output Jacobian statistics show the same pattern as in our MLP and CNN experiments: AdamO preserves substantially higher effective rank and lower condition number, indicating that gradient propagation remains richer and less anisotropic over continual training. For the weight and dormant-neuron diagnostics, we show the skip-connection transformer runs, where the same pattern persists: AdamO maintains broader effective rank, tighter control of the smallest and largest singular values, and better preservation of the overall singular-value scale, while

*Table 2.* Octax Continual Learning: Environment and Hyperparameters

| Parameter | Value |
|---|---|
| *Environment* | |
| Observation shape | $(4, 32, 64)$ |
| Action space | 6 (unified) |
| Games | 8 |
| Cycles | 3 |
| Steps per task | $5 \times 10^6$ |
| *Network Architecture* | |
| CNN Layer 1 | 32 filters, $8 \times 4$ kernel, stride $(4, 2)$ |
| CNN Layer 2 | 64 filters, $4 \times 4$ kernel, stride $(2, 2)$ |
| CNN Layer 3 | 64 filters, $3 \times 3$ kernel, stride $(1, 1)$ |
| MLP hidden layers | $(64, 64, 64, 64)$ |
| Activation | ReLU |
| *PPO Hyperparameters* | |
| Parallel environments | 512 |
| Rollout length | 32 |
| Epochs per update | 4 |
| Minibatches | 32 |
| Learning rate | $5 \times 10^{-4}$ |
| Discount ($\gamma$) | 0.99 |
| GAE $\lambda$ | 0.95 |
| Clip $\epsilon$ | 0.2 |
| Entropy coefficient | 0.01 |
| Value function coefficient | 0.5 |
| Max gradient norm | 0.5 |

also reducing feature collapse. Taken together, these figures support the claim that the dynamical-isometry perspective remains useful in attention-based architectures, while also reinforcing that a comprehensive study on larger transformers is still needed.

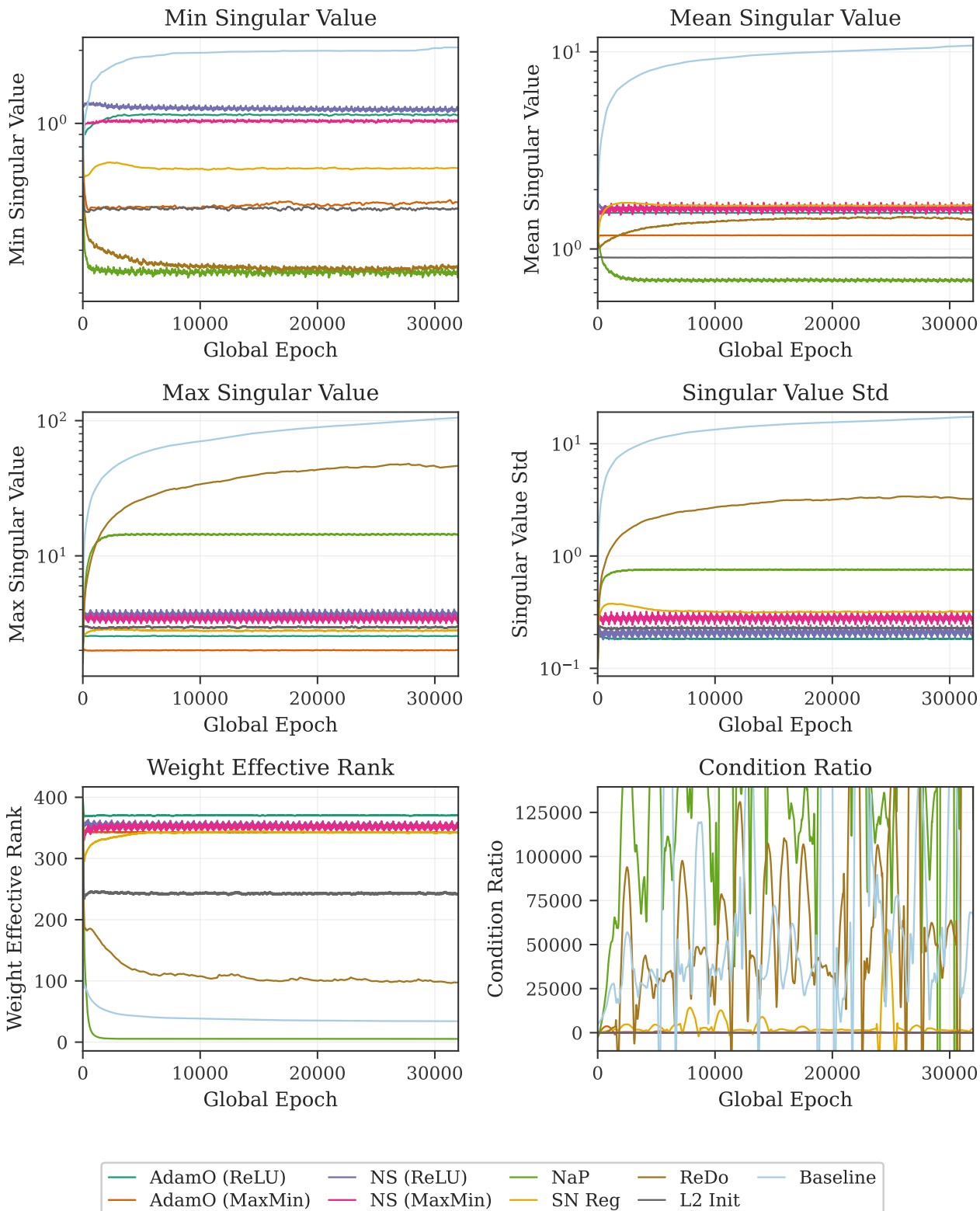

*Figure 4.* Weight-space diagnostics for CIFAR-10 pixel permutation, including singular-value statistics, effective rank, and the weight condition ratio over training.

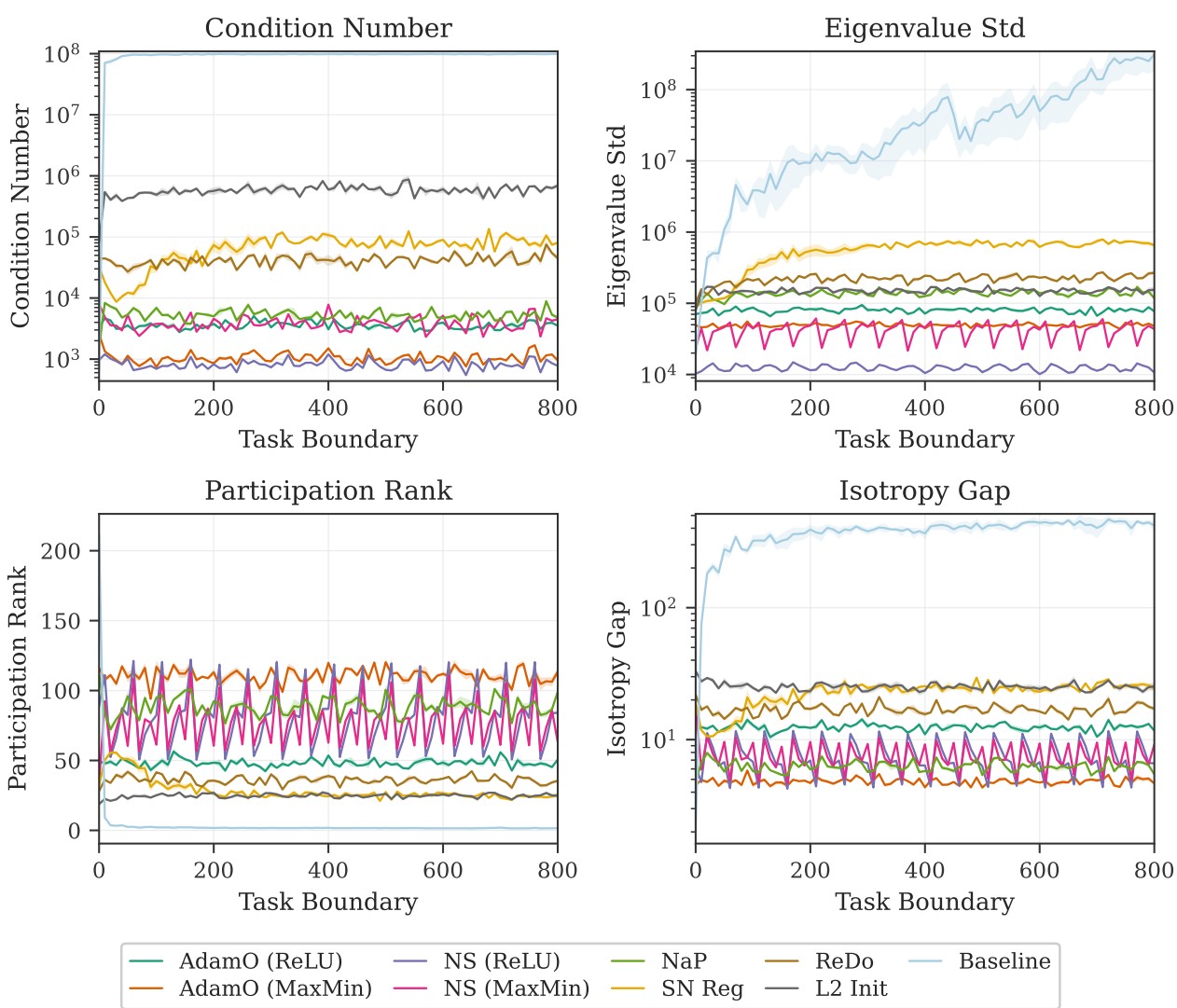

*Figure 5.* Core empirical NTK diagnostics for CIFAR-10 pixel permutation, including condition number, eigenvalue spread, participation rank, and isotropy gap.

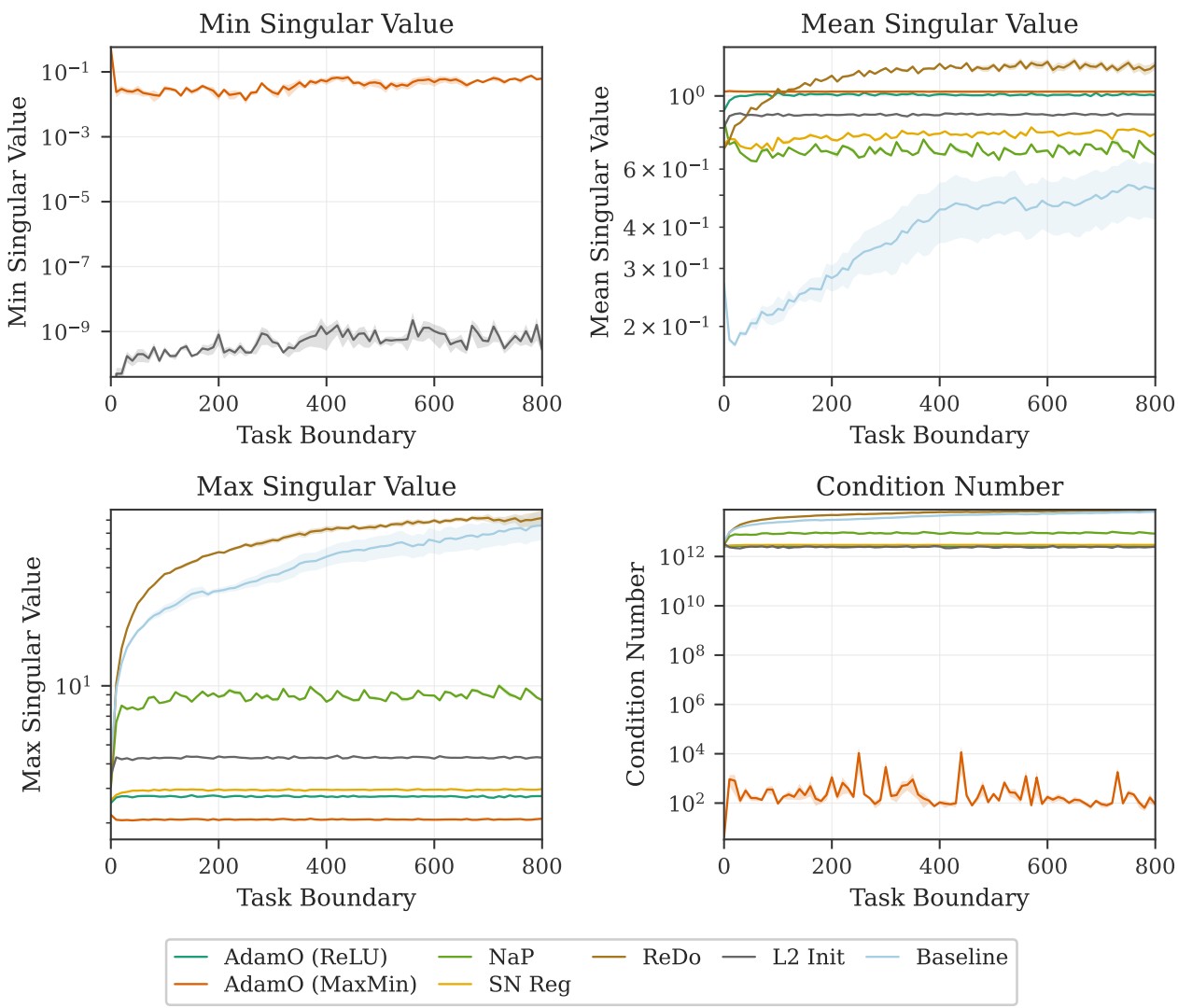

*Figure 6.* Intermediate-layer Jacobian diagnostics for CIFAR-10 pixel permutation, showing how the singular-value spectrum and conditioning evolve through training.

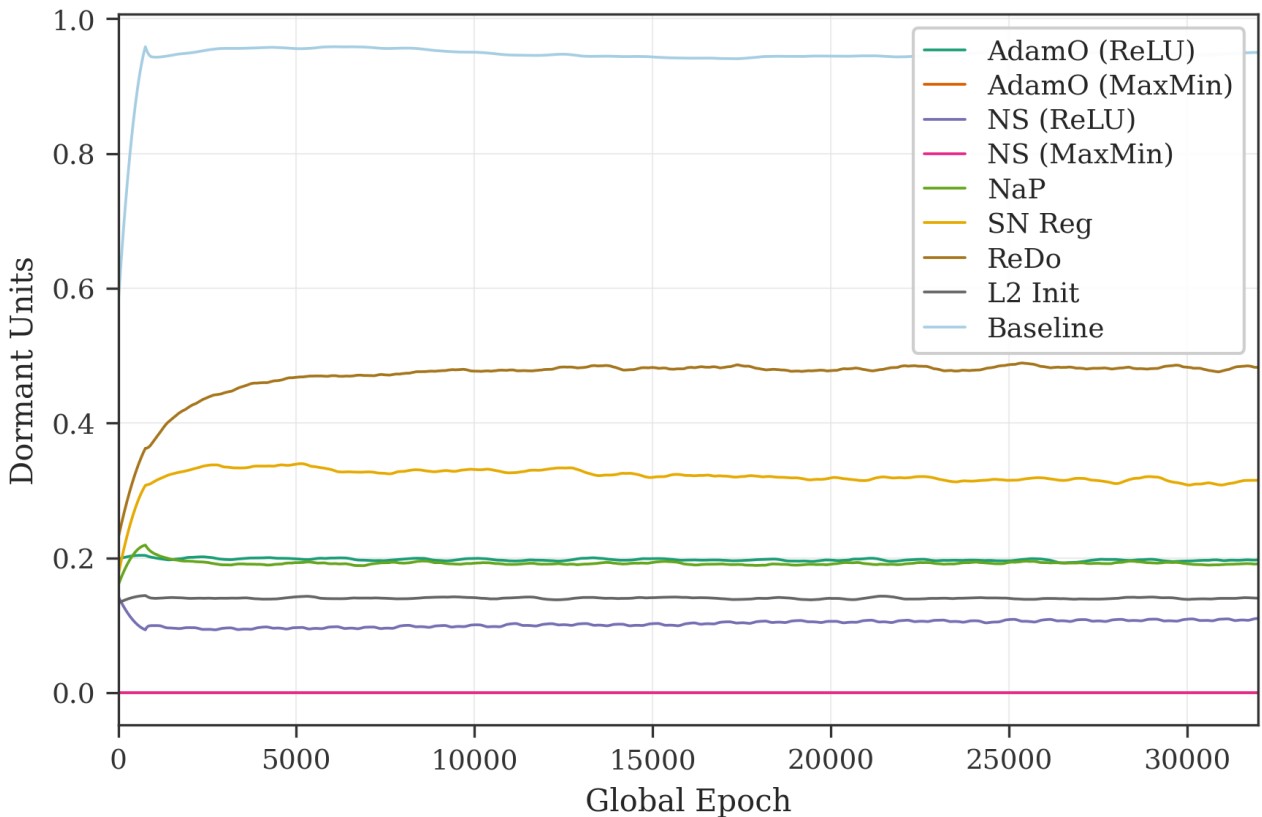

*Figure 7.* Dormant-unit diagnostics for the supervised continual-learning experiments. The figure shows how the number of Sokar-style dormant ReLU units evolves during training, and how the isometry-preserving methods reduce or reverse the buildup of inactive features relative to the baselines.

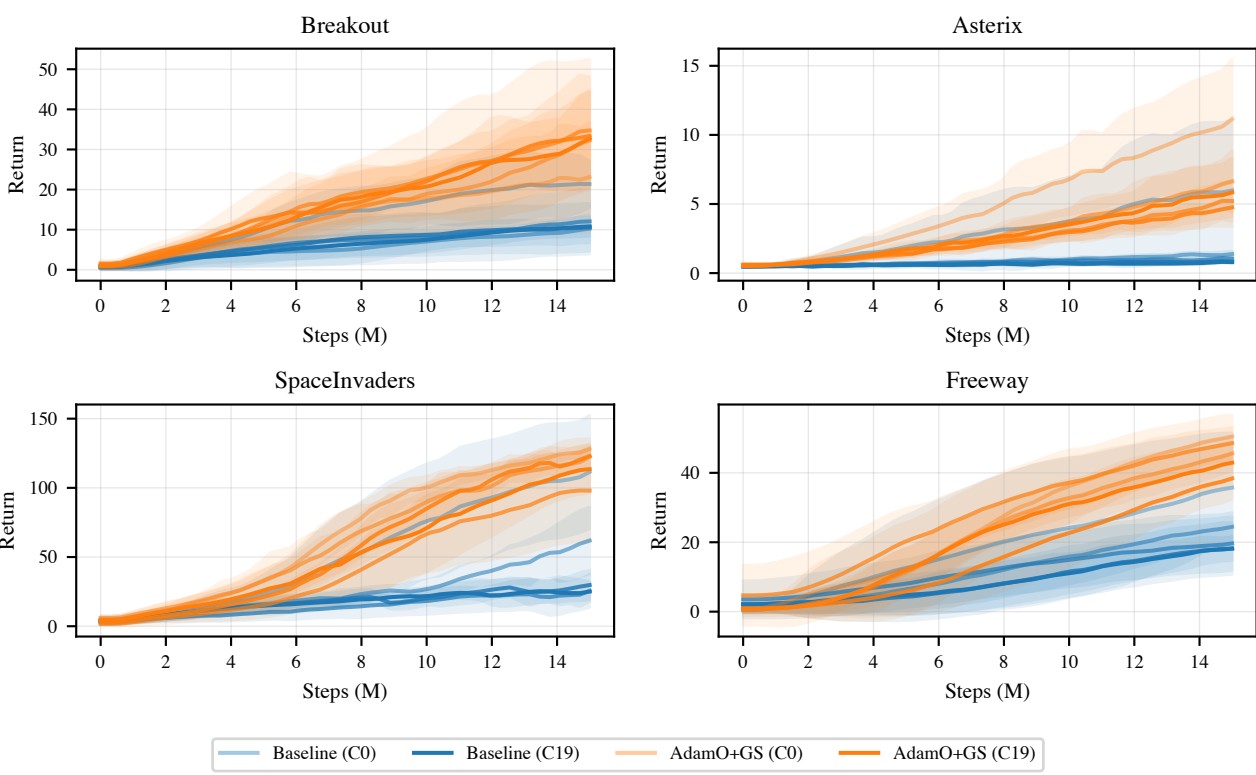

*Figure 8.* Training curves for MinAtar games. The color gradient indicates early (light) to late (dark) cycles (20 cycles).

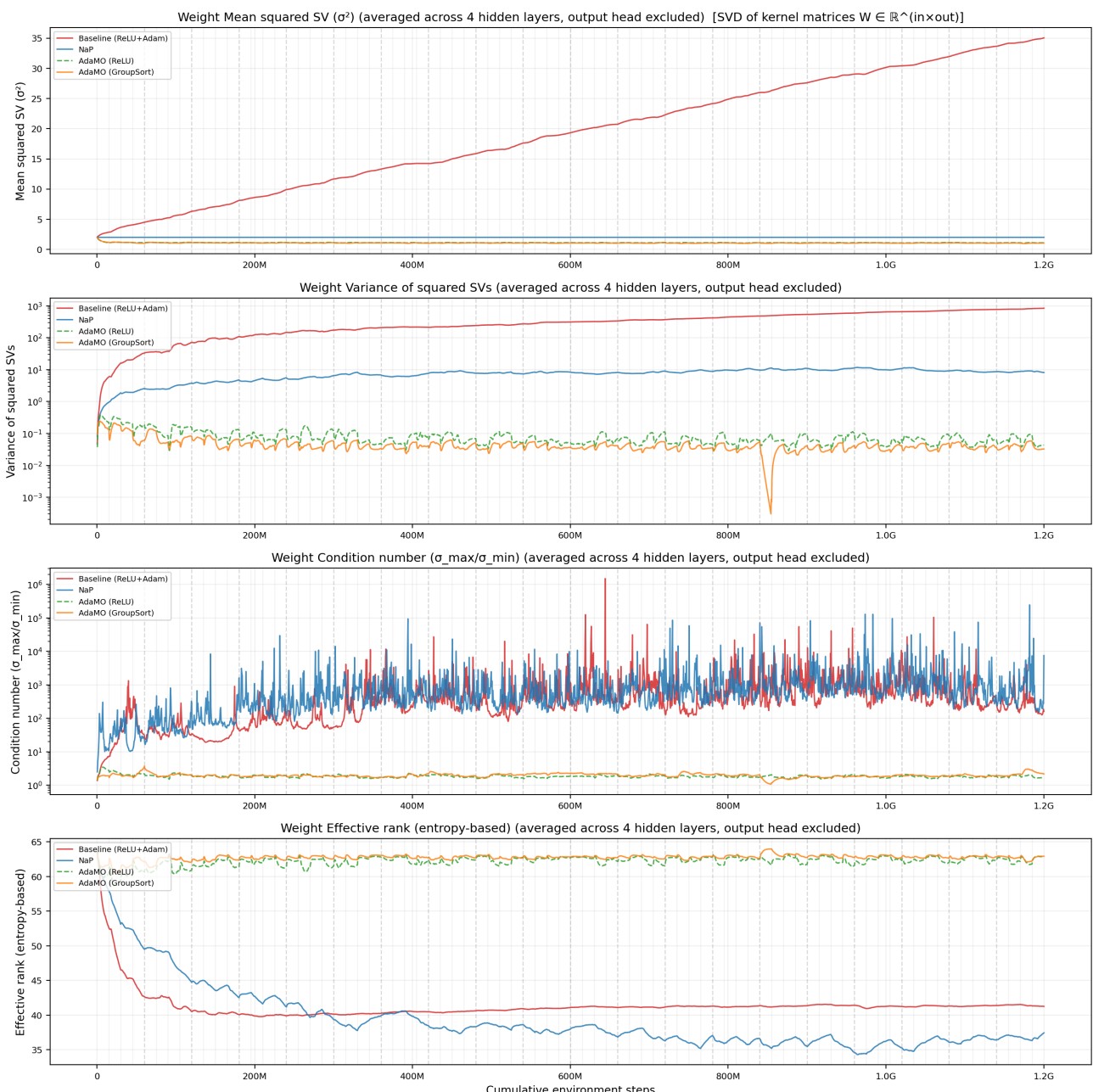

*Figure 9.* Weight-spectrum diagnostics for continual MinAtar. The plotted quantities summarize the singular-value distribution of the learned weight operators over training, making visible whether layers develop strong anisotropic directions or preserve a tighter spectrum.

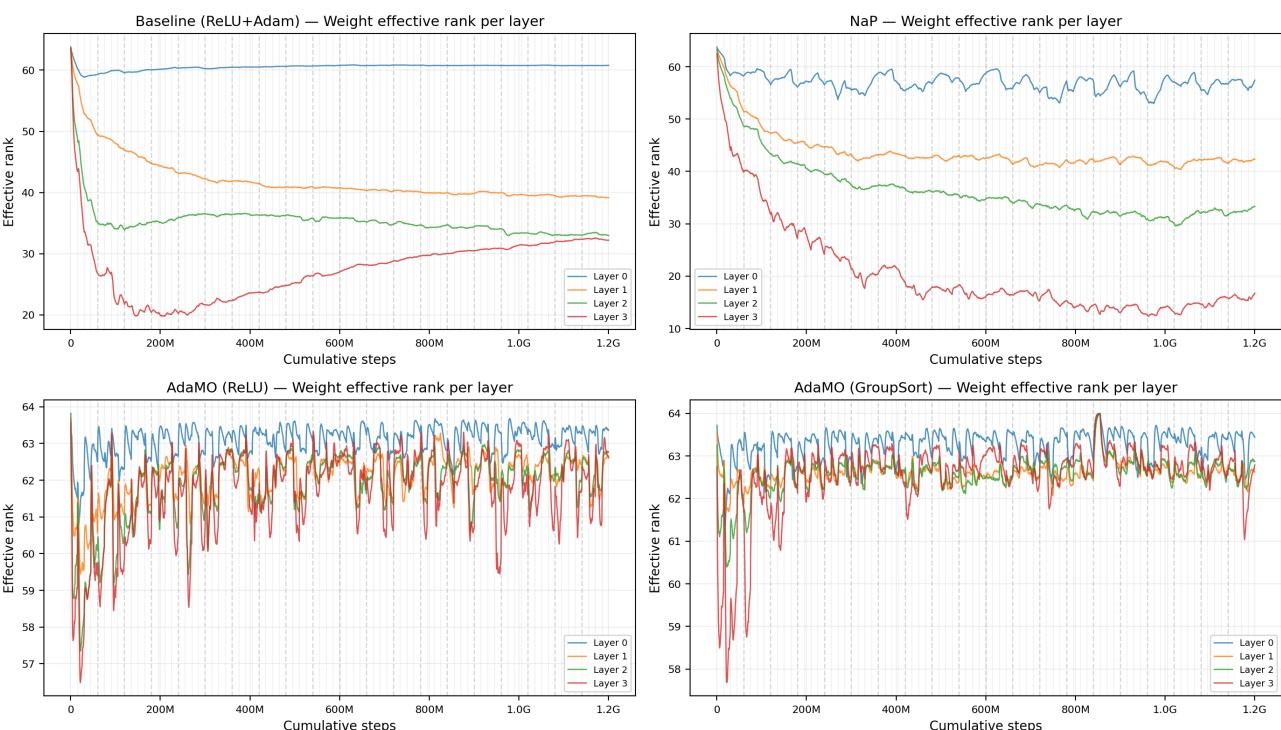

*Figure 10.* Layer-wise weight effective-rank diagnostics for continual MinAtar. These panels track how many singular directions of each layer remain meaningfully used, helping distinguish balanced representations from spectra that collapse onto a small subspace.

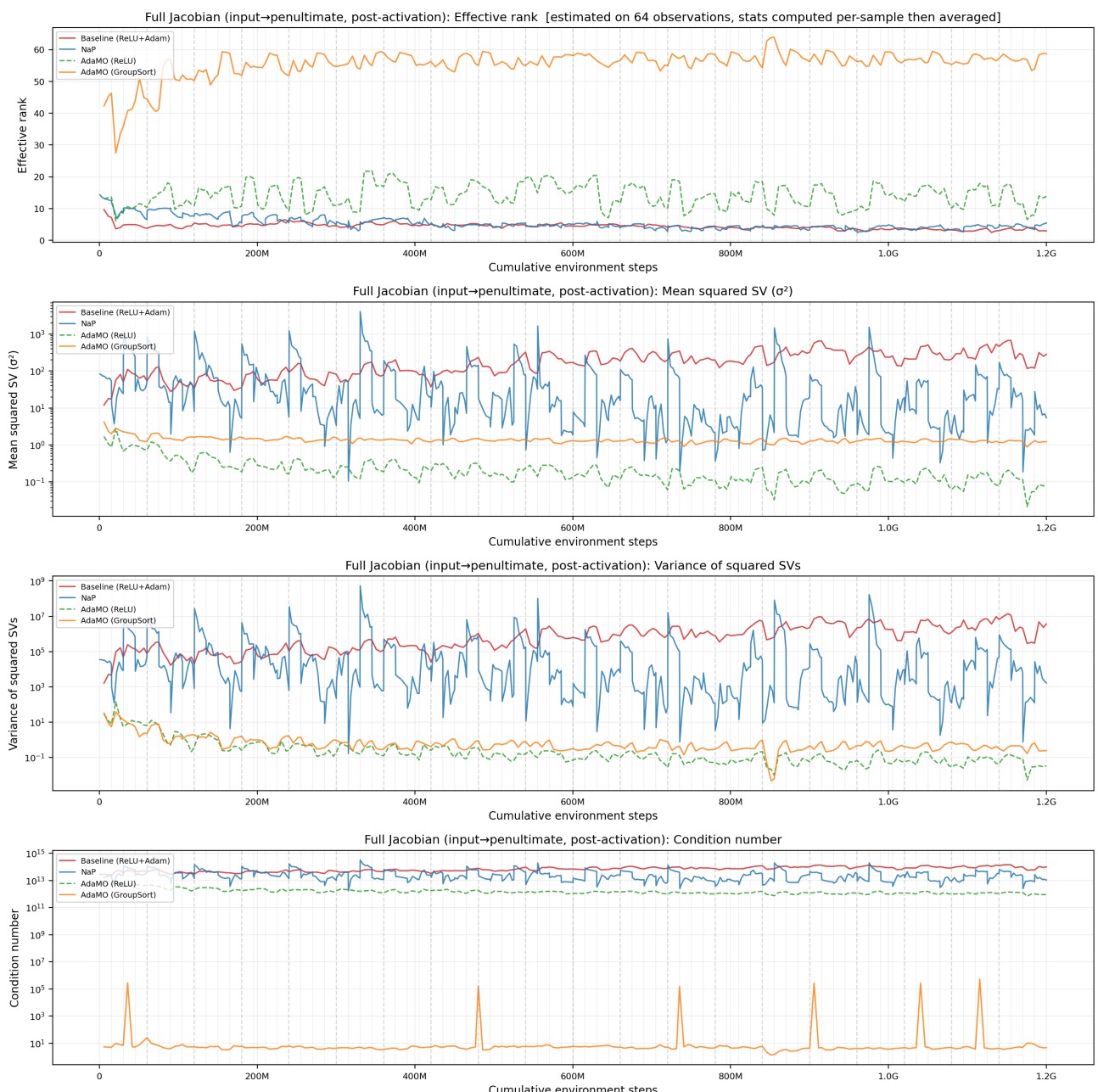

*Figure 11.* Full Jacobian diagnostics for continual MinAtar. These figures summarize the singular-value spectrum of the input-output Jacobian, directly probing dynamical isometry through quantities such as singular-value spread and conditioning.

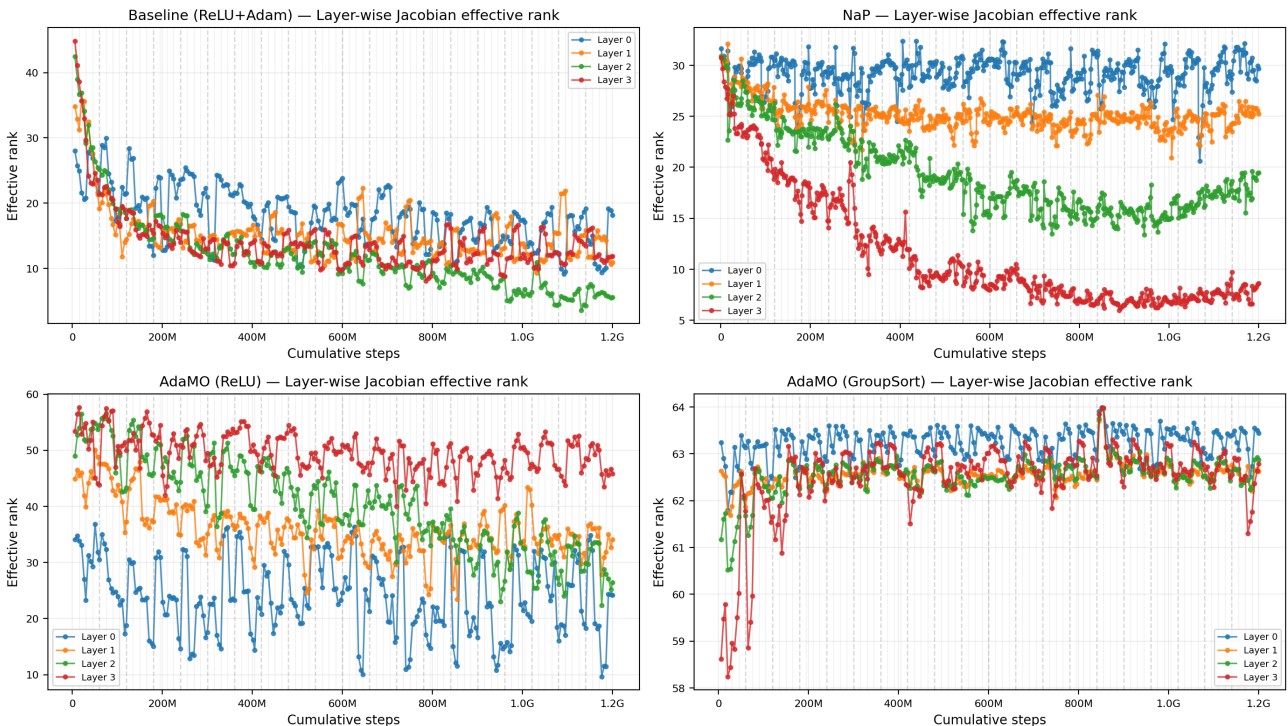

*Figure 12.* Layer-wise Jacobian diagnostics for continual MinAtar. Unlike the full Jacobian view, these panels localize where along the network depth singular values drift away from one, revealing which layers are responsible for deteriorating gradient transport.

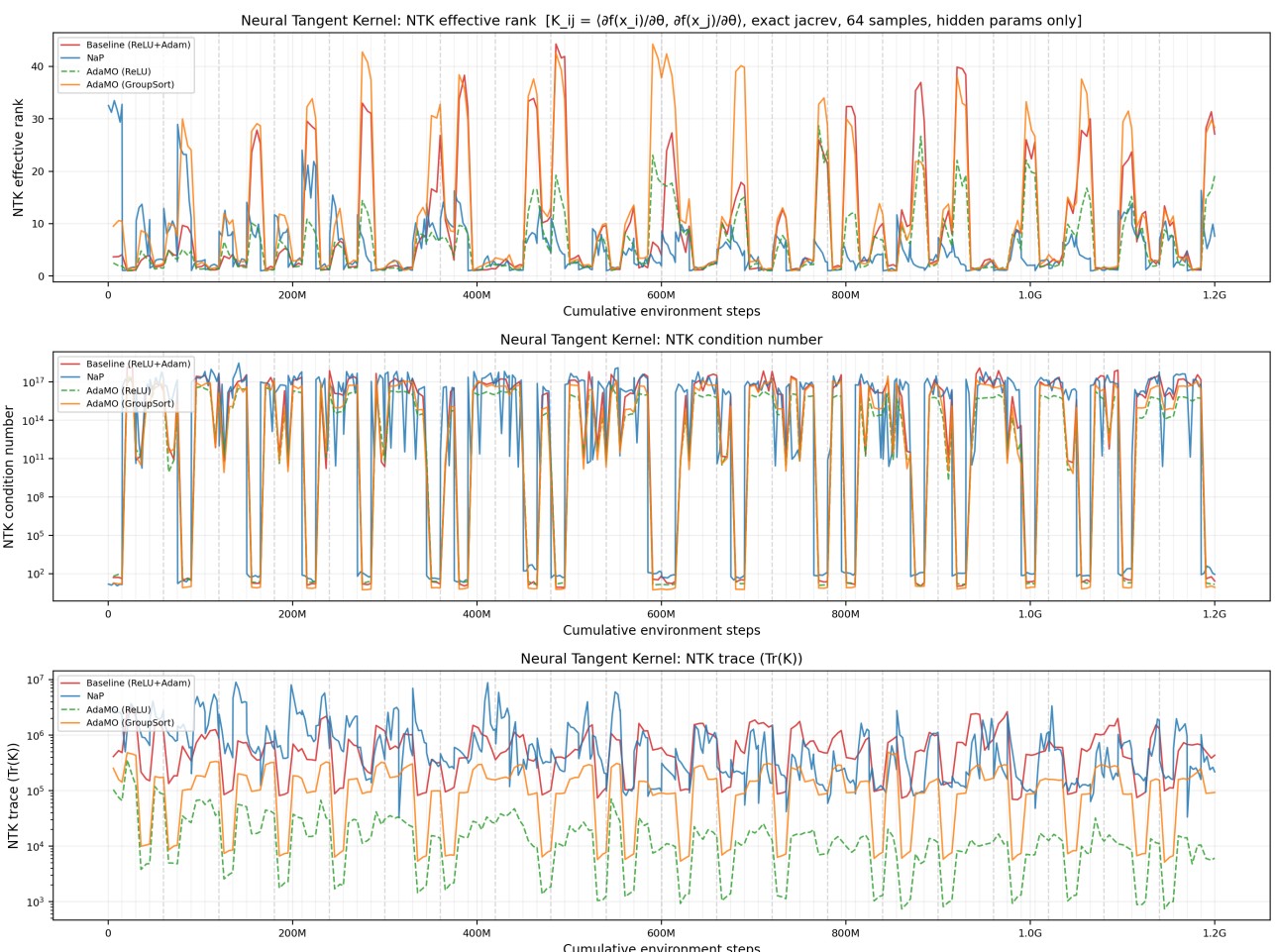

*Figure 13.* Empirical NTK diagnostics for continual MinAtar. These panels track kernel conditioning and rank-related quantities, indicating how isotropically parameter updates can move the represented function in output space over time.

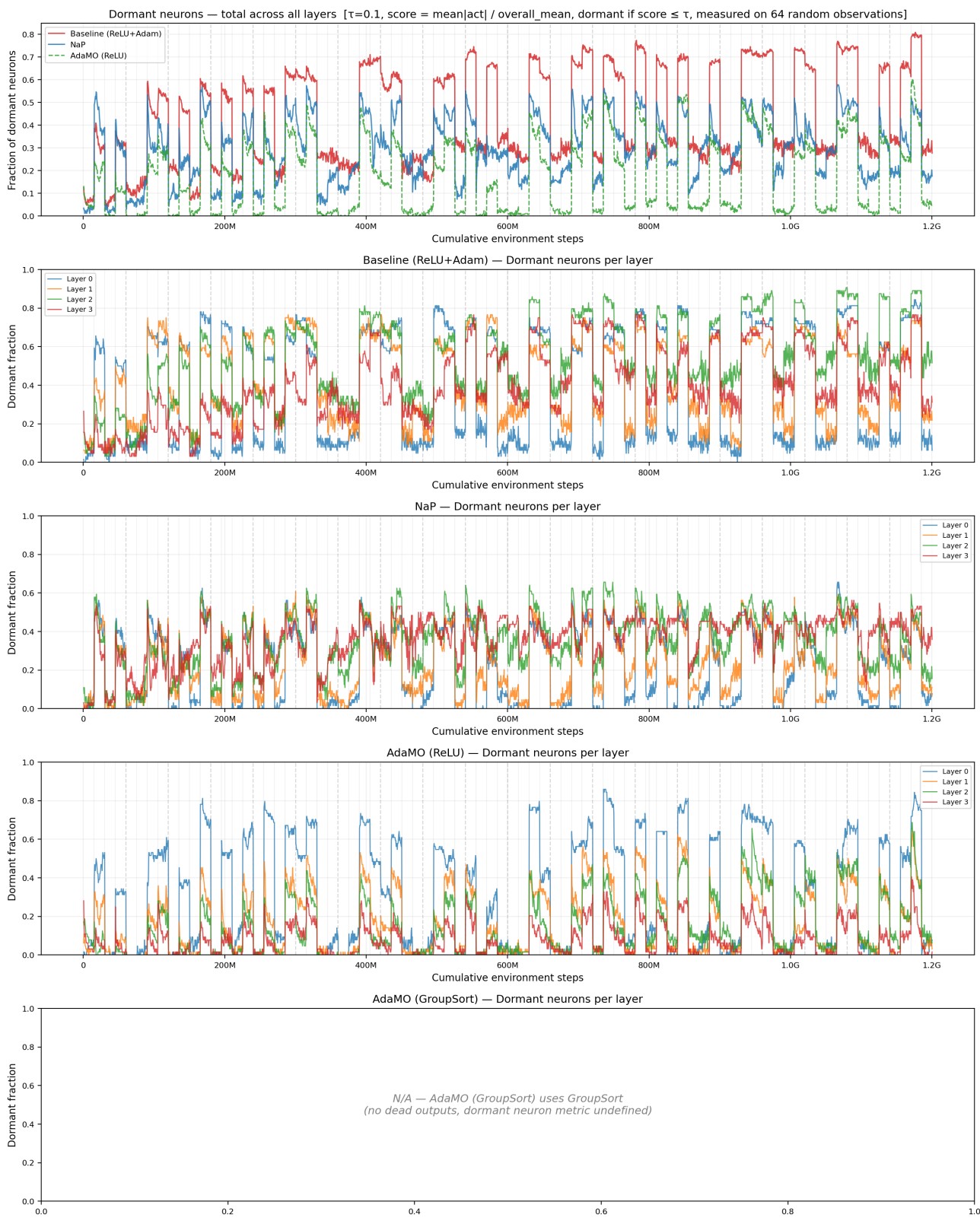

*Figure 14.* Dormant-neuron diagnostics for continual MinAtar. These figures quantify inactive or weakly active units, making the connection between revival of dormant features and preserved plasticity explicit.

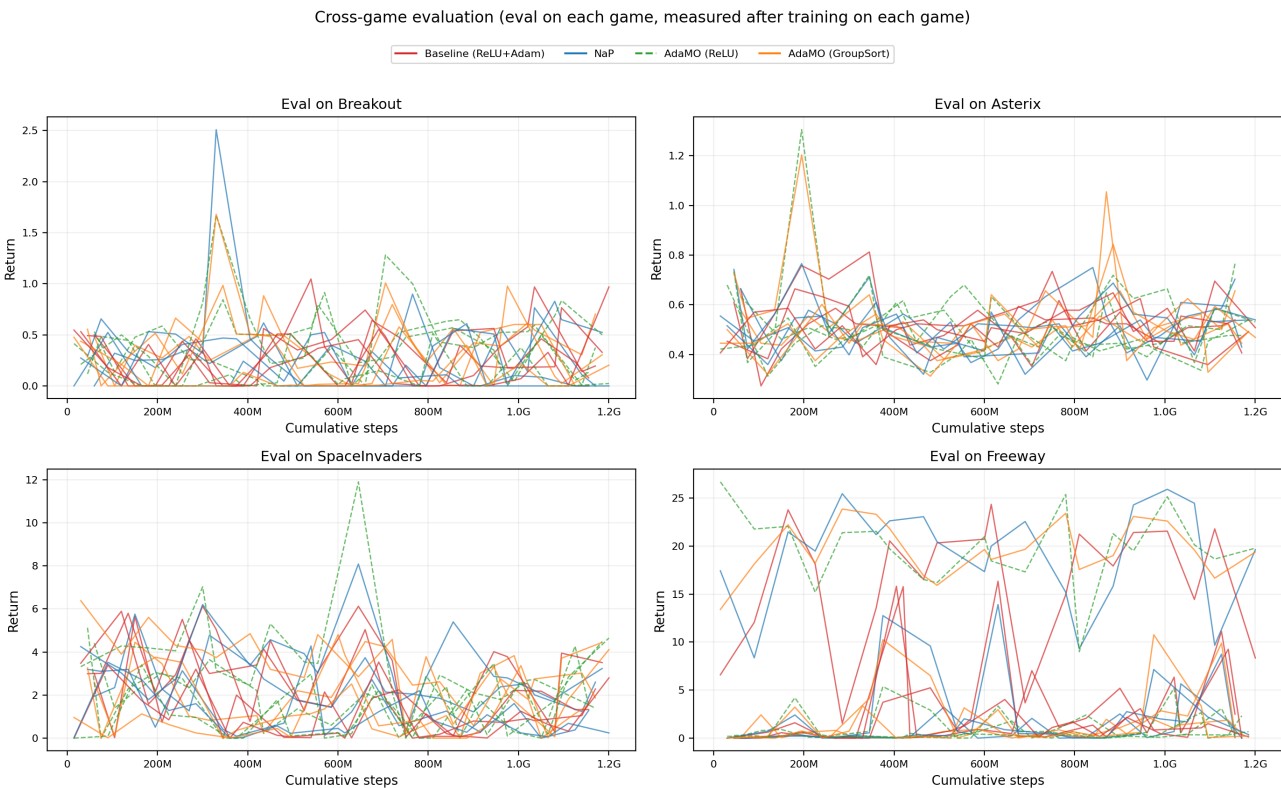

*Figure 15.* Per-game evaluation performance for continual MinAtar. Breaking the aggregate score down by environment shows whether gains come from broadly preserved plasticity across games rather than from improvements on only a small subset.

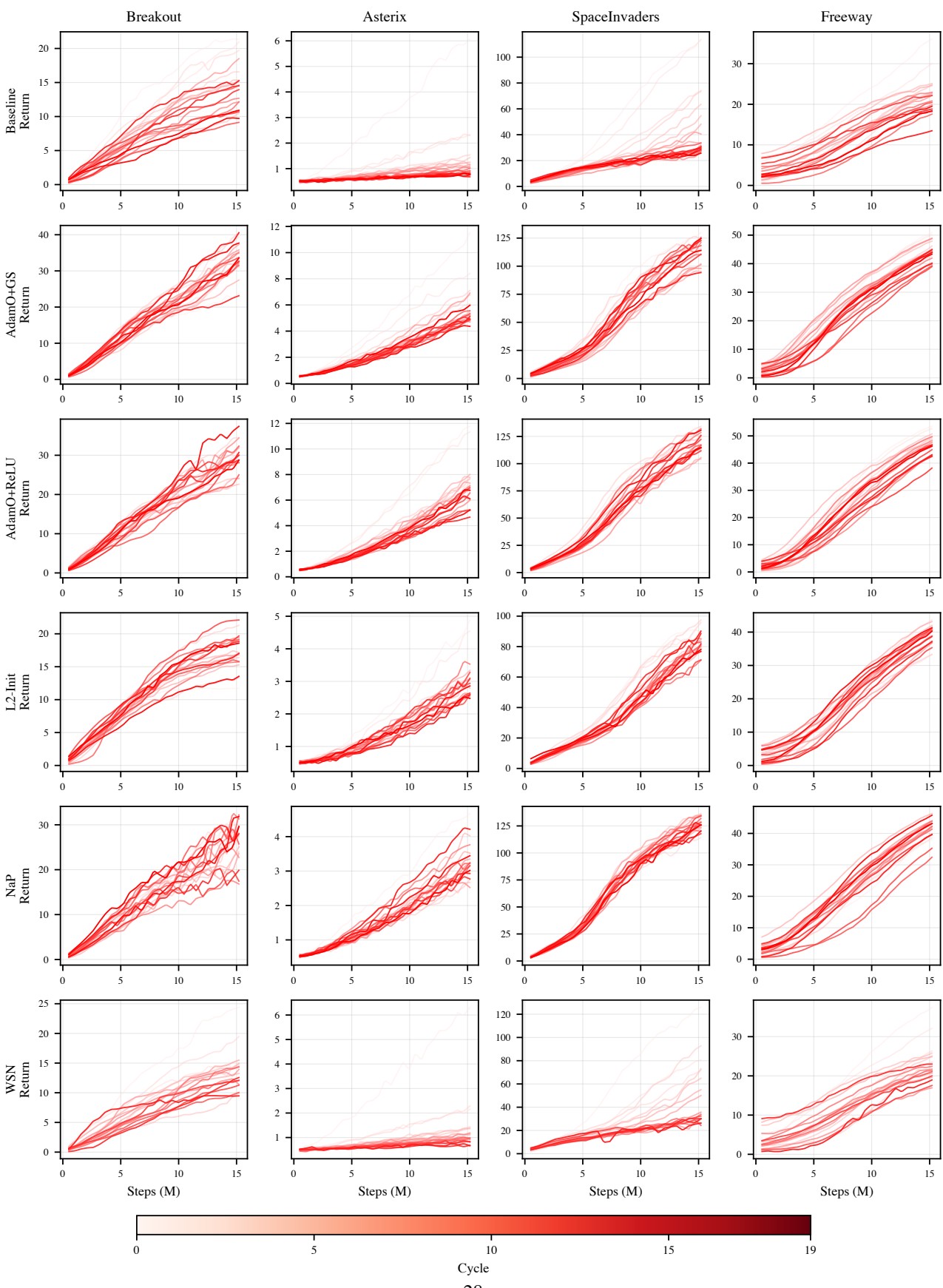

*Figure 16.* Training curves for MinAtar games. The color gradient indicates early (light) to late (dark) cycles (20 cycles).

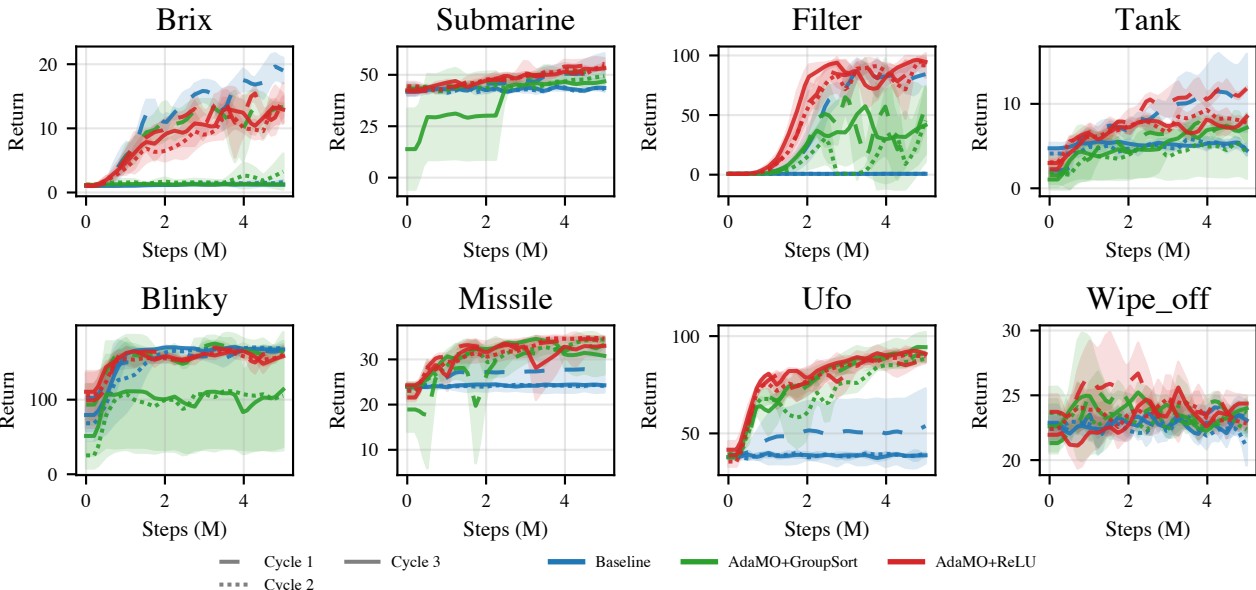

*Figure 17.* Training curves for Octax games showing return on evaluation environments against training steps. The line style indicates the cycle within the continual experiment.

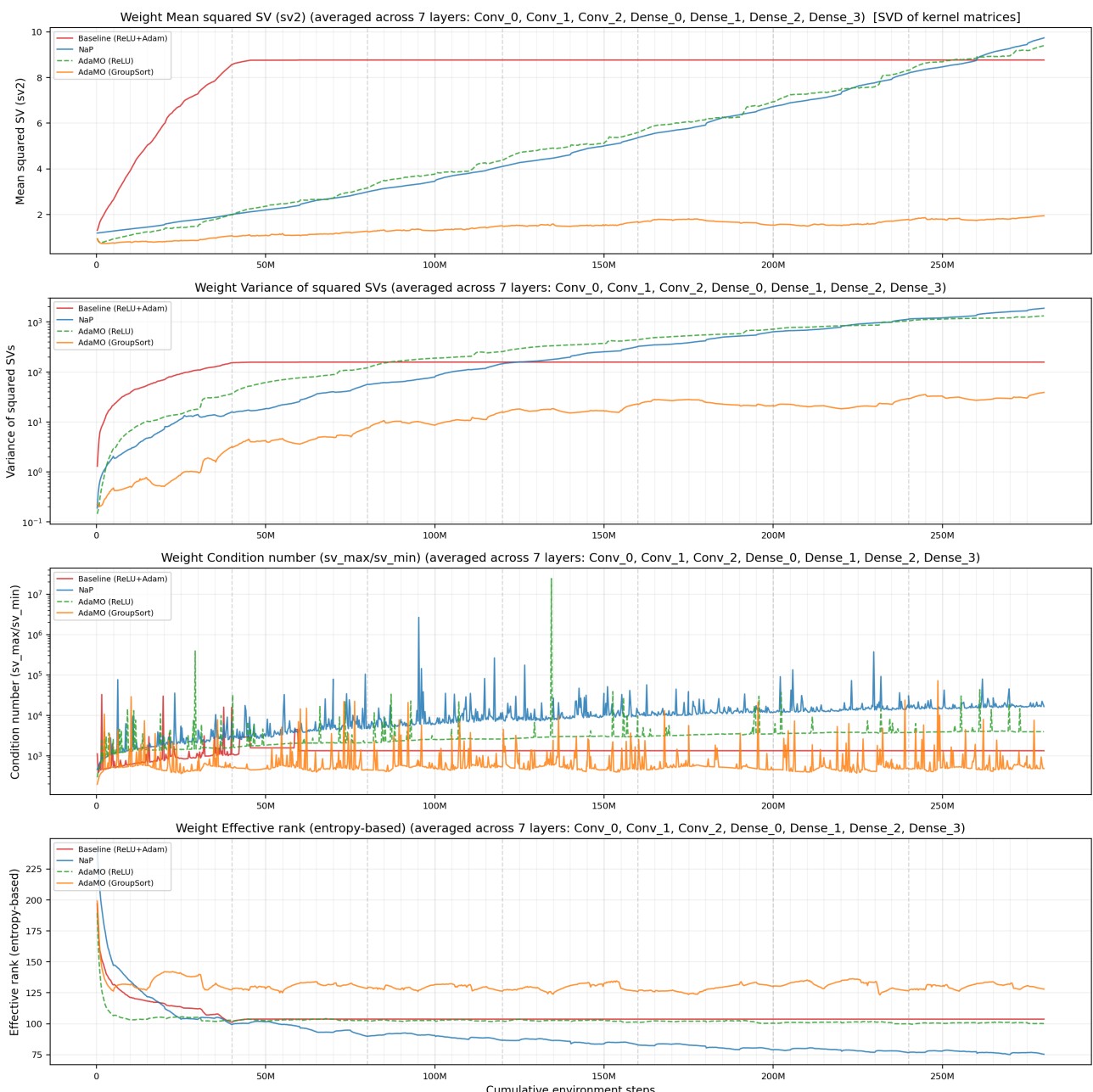

*Figure 18.* Weight-spectrum diagnostics for continual Octax. These panels summarize the singular-value statistics of the convolutional and linear operators, highlighting whether training preserves a balanced spectrum or develops highly anisotropic directions.

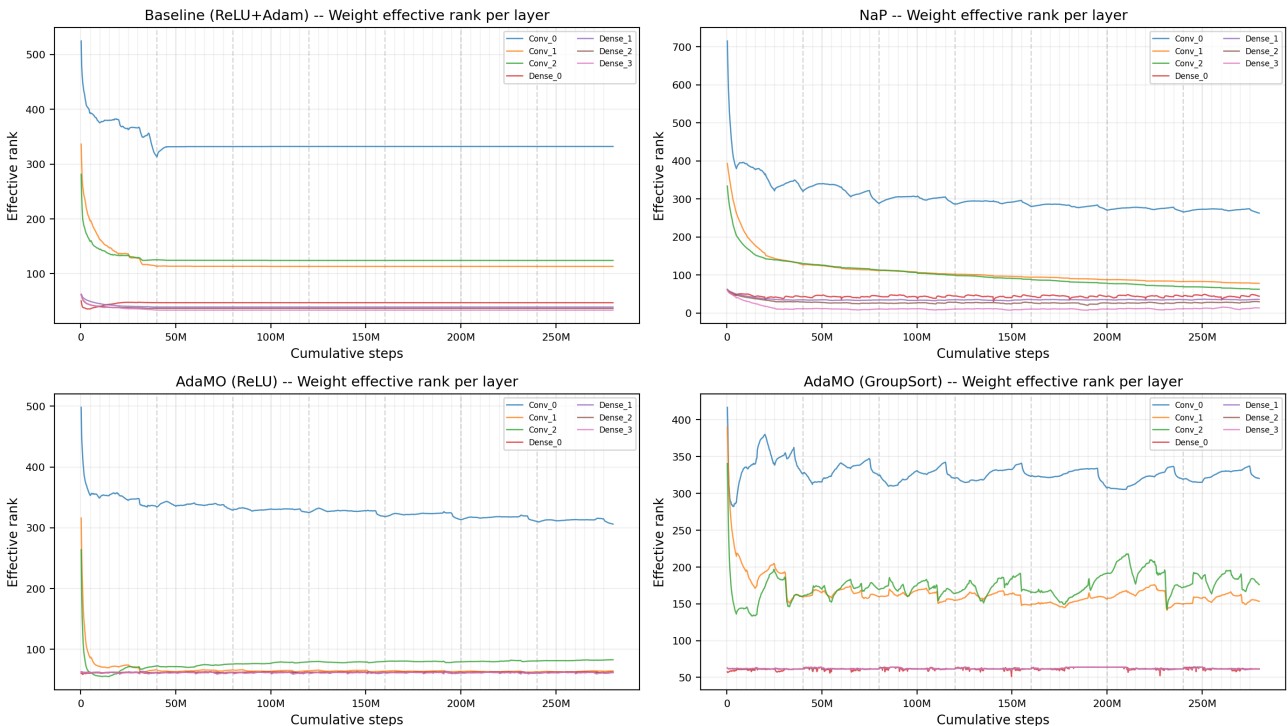

*Figure 19.* Layer-wise weight effective-rank diagnostics for continual Octax. Effective rank measures how broadly each layer uses its singular directions, complementing raw norm or spectral diagnostics with a notion of dimensional richness.

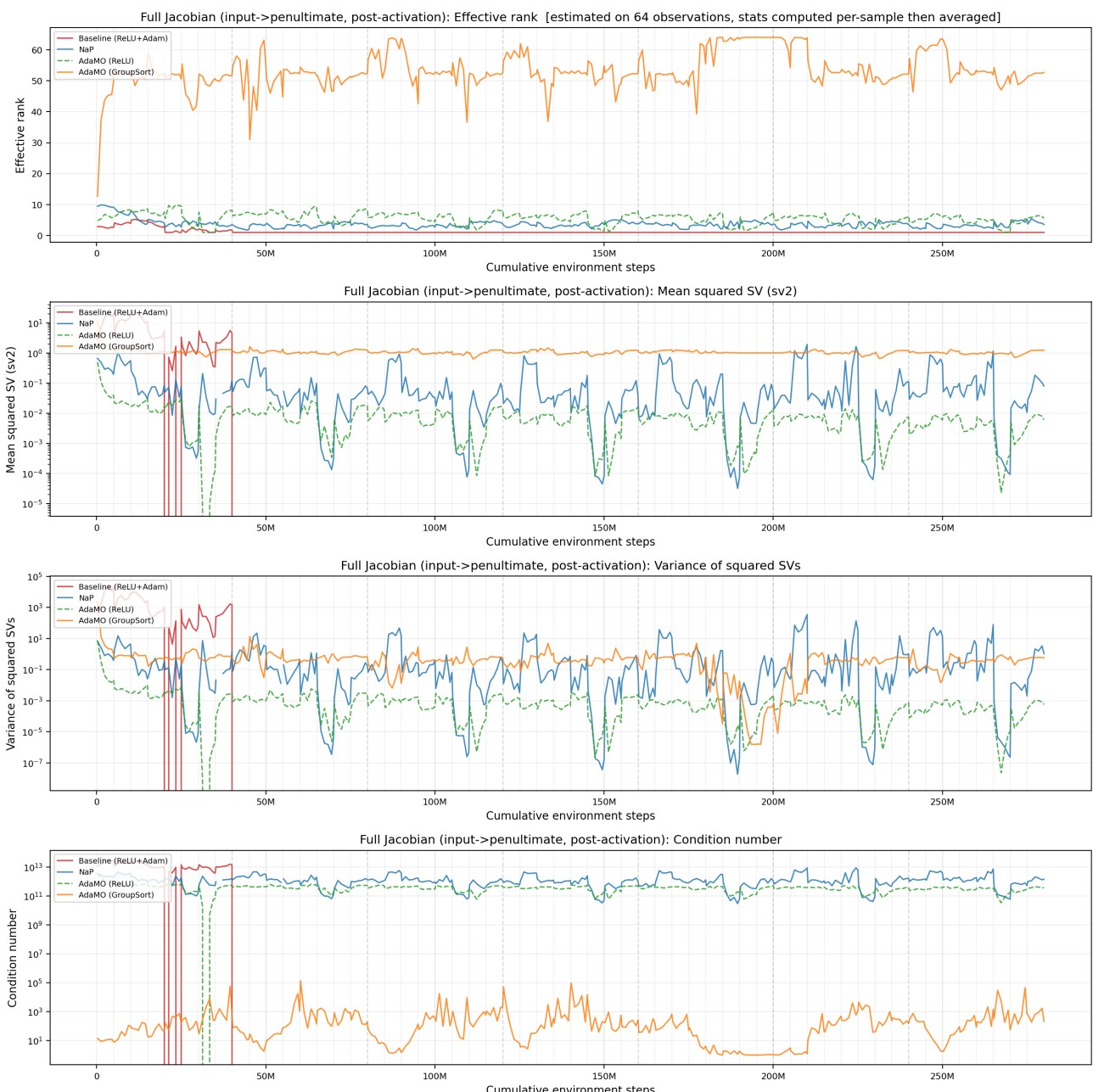

*Figure 20.* Full Jacobian diagnostics for continual Octax. These plots summarize the singular-value spectrum of the end-to-end Jacobian and therefore directly monitor whether the network stays near a dynamically isometric regime.

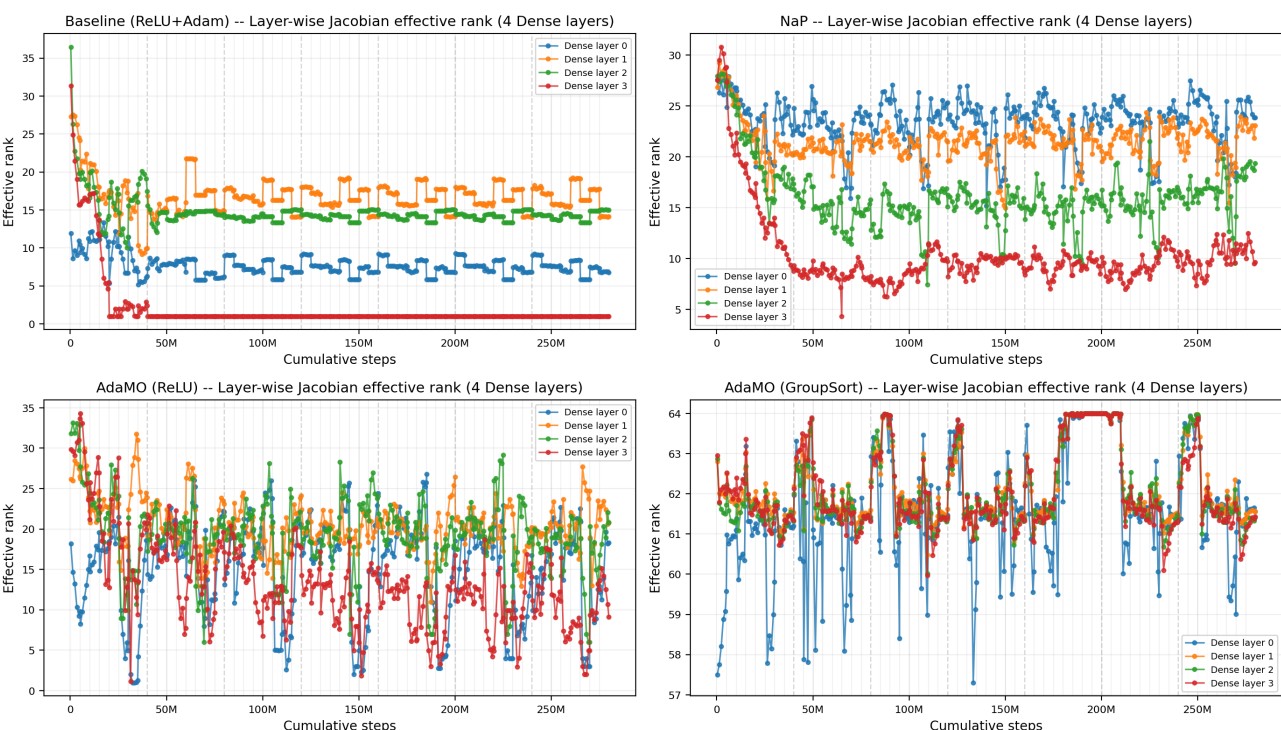

*Figure 21.* Layer-wise Jacobian diagnostics for continual Octax. By decomposing Jacobian statistics across depth, these panels identify where conditioning degrades and where isometry-preserving methods stabilize signal propagation.

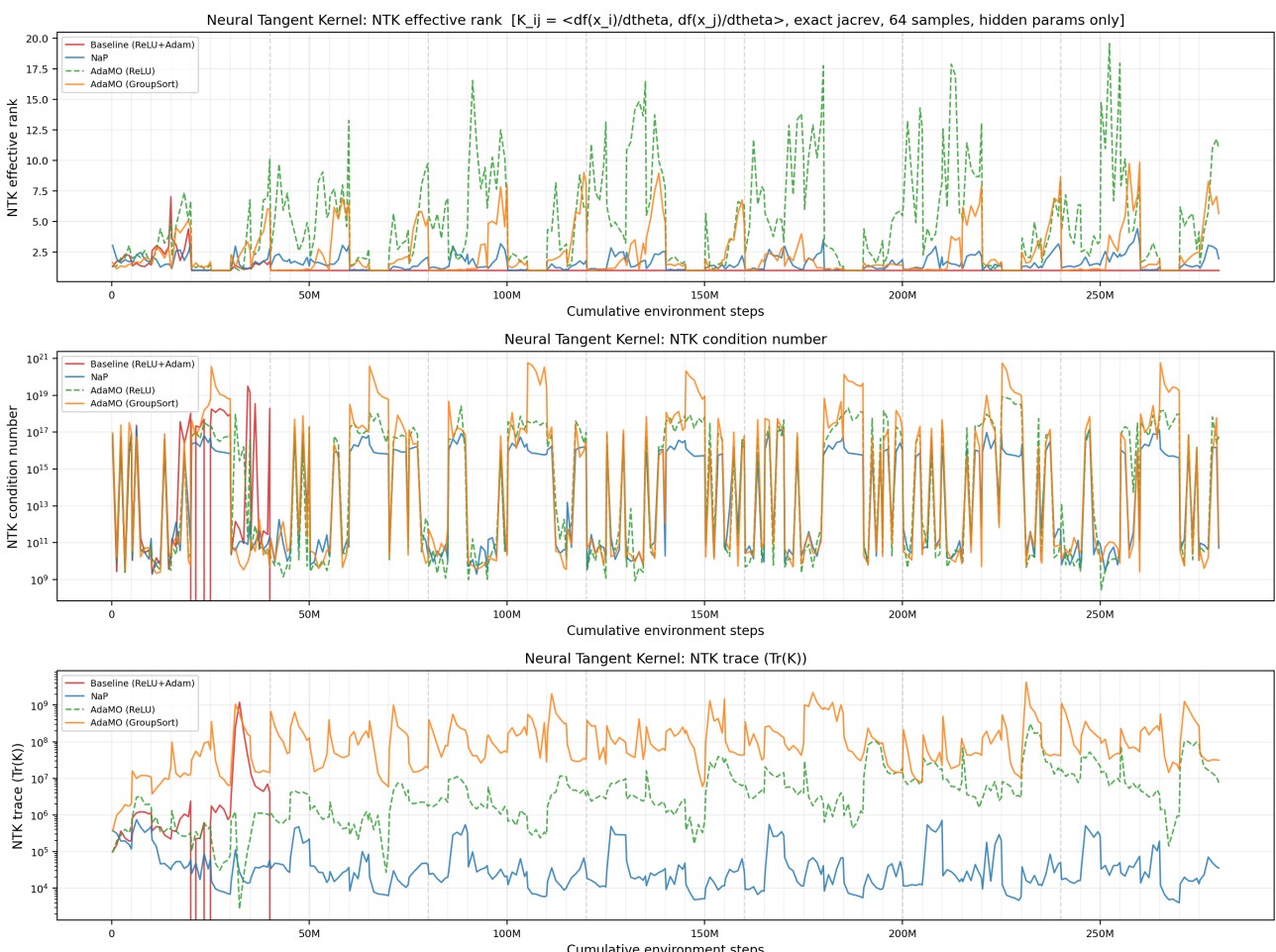

*Figure 22.* Empirical NTK diagnostics for continual Octax. These figures track kernel condition numbers and rank-related measures, which quantify how uniformly policy/value gradients can move the represented function.

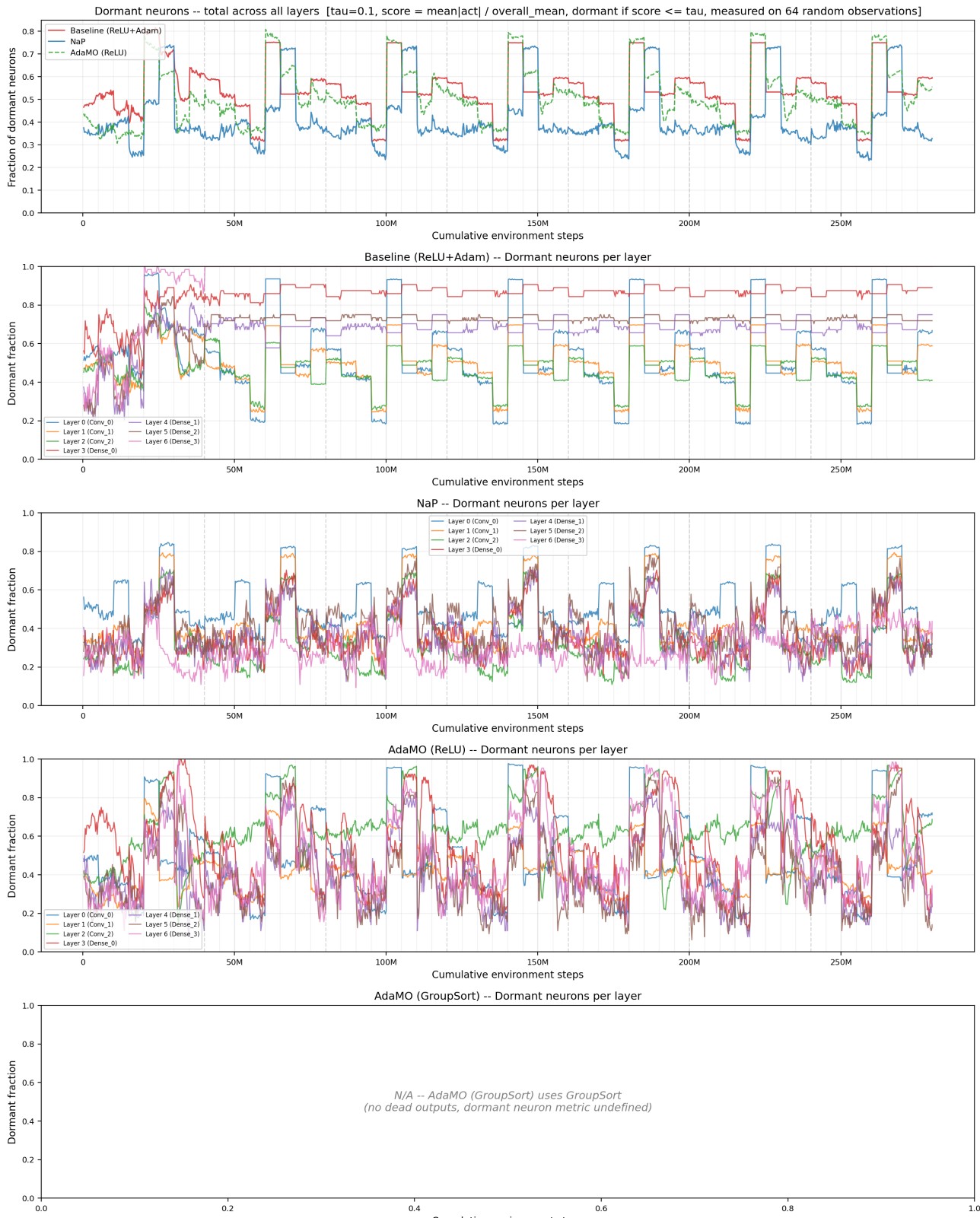

*Figure 23.* Dormant-neuron diagnostics for continual Octax. These panels measure inactivity and feature collapse in the network, providing an RL-side analogue of the dormant-unit behavior discussed for supervised settings.

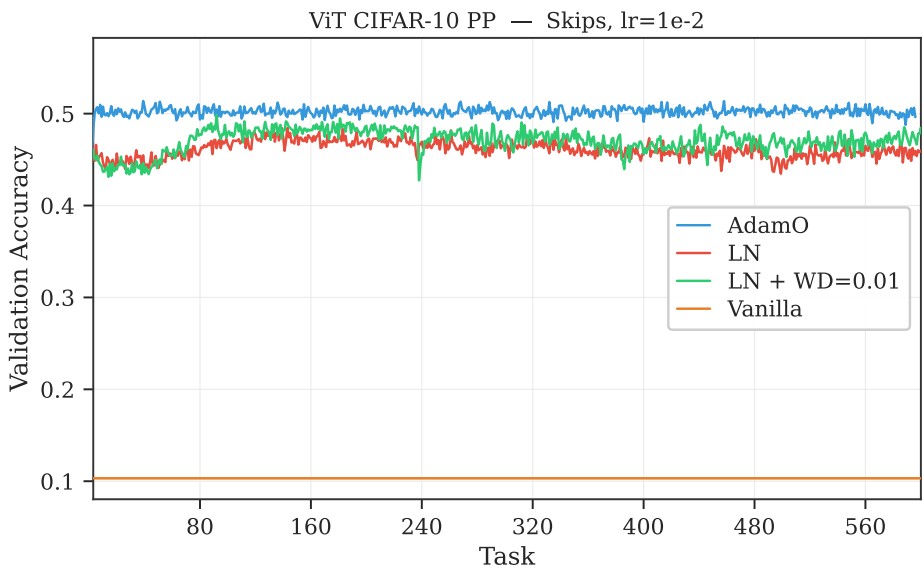

*Figure 24.* Validation accuracy for the continual CIFAR-10 pixel-permutation transformer experiments at learning rate $10^{-2}$. This plot summarizes the primary stability and performance comparison across the transformer variants considered in the rebuttal study.

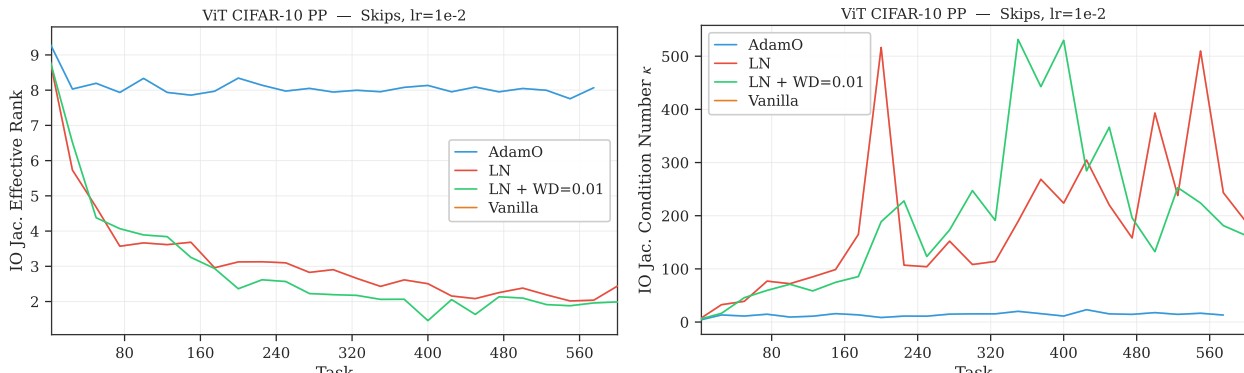

*Figure 25.* Input-output Jacobian diagnostics for the transformer experiments with skip connections at learning rate $10^{-2}$. Left: input-output effective rank, which measures how many singular directions of the end-to-end Jacobian remain meaningfully used. Right: input-output condition number, which captures worst-case anisotropy of gradient transport. Together these panels directly probe whether the transformer remains in a healthier dynamical-isometry regime throughout continual learning.

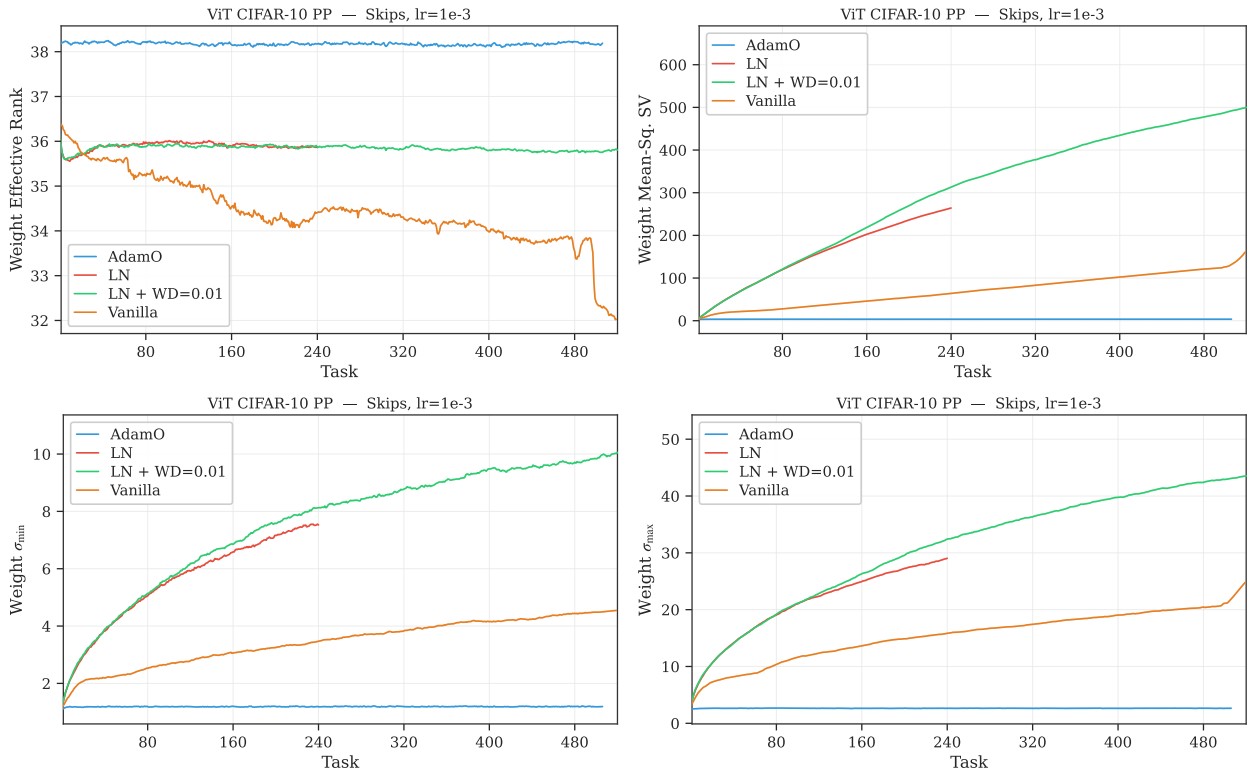

*Figure 26.* Weight-spectrum diagnostics for the transformer experiments with skip connections. The effective-rank panel measures how broadly each layer uses its singular directions; the mean-squared singular-value panel tracks preservation of the overall weight scale; and the smallest and largest singular-value panels expose anisotropic extremes. These figures make clear whether the optimizer preserves a balanced spectrum rather than allowing progressive spectral collapse.

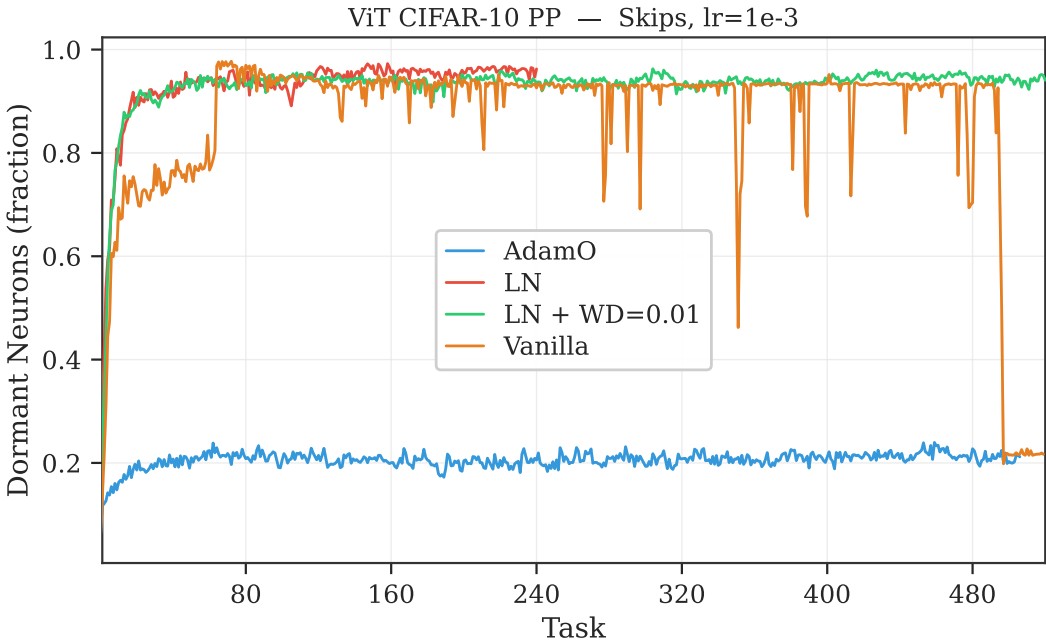

*Figure 27.* Dormant-neuron diagnostics for the transformer experiments with skip connections. This plot tracks the buildup of inactive or weakly active units over training and shows whether improved conditioning is accompanied by reduced feature collapse.

