# OpenReview forum: "Preserving Plasticity in Continual Learning via Dynamical Isometry"
_ICML.cc/2026/Conference — ICML 2026 regular_

### Official Review · Reviewer_1wUh · 2026-03-10

**Soundness:** 3
**Presentation:** 3
**Significance:** 2
**Originality:** 3
**Overall Recommendation:** 5
**Confidence:** 3

**Summary:**

This paper studies the loss of network plasticity during continual training by introducing plasticity. The authors identify the deviation of layerwise Jacobian singular values from 1, as the main cause of plasticity loss. They provide theoretical results showing that almostly isometric networks can preserve plasticity. Experiments on supervised and RL tasks with MLPs and CNNs suggest that the proposed approach effectively preserves plasticity and outperforms baseline methods.

**Compliance With Llm Reviewing Policy:**

Affirmed.

**Final Justification:**

The author have solved my concerns and questions, and the AdamO method is proved to have better stability in training empirically, so I have raised my score to Acceptance.

**Key Questions For Authors:**

Besides the weakeness,
1. How about the training stability and efficiency after introducing AdamO?
2.  Is AdamO better than baselines in maintaining isometry, with quantitative performance?

**Limitations:**

yes

**Strengths And Weaknesses:**

Strengths
1.	The paper points out that the loss of plasticity is caused by the departure from dynamical isometry, which means the layer-wise Jacobian singular values drift away from 1. It also builds a unified theoretical framework with strict derivation.
2.	Based on layer-wise dynamical isometry, it designs the AdamO optimizer. This optimizer is tested in different training scenarios and performs better.

Weakness

1.	The authors only test traditional architectures such as MLP and CNN. It does not extend to mainstream architectures like Transformer, nor does it cover LLM fine-tuning scenarios.
2.	The authors do not conduct in-depth analysis on the internal mechanism of the method’s performance. For example, it does not verify whether AdamO can better maintain dynamical isometry. Meanwhile, it fails to prove that the proposed layer-wise dynamical isometry and AdamO are sufficient conditions for keeping plasticity.

---

> ### Author Rebuttal · Authors · 2026-03-31
>
> Thank you for taking the time to engage with our paper, and for highlighting our contributions and the performance of our optimizer. We hope the following points address your questions:
>
> **On Transformers**
> Our paper emphasizes the importance of both well-conditioned weights and activations. At the same time, attention is often a (highly) contractive, and thus non-isometric, operation. This is a key reason larger transformers rely on skip connections, which improve gradient propagation outside the residual branches (though they in turn require normalization for stability because the singular values of the skip and residual paths interact). Given these differences, we considered a full isometry-based analysis of transformers beyond the scope of this work and listed it as future work in the Discussion. That said, we are attempting to set up small-scale transformer experiments inspired by Osband’s recent work on Delightful Policy Gradient to study orthogonal regularization and its interaction with attention. We plan to still include these as a preliminary analysis in the paper, while leaving a thorough large-scale study to future work.
>
> **Quantitative extension of diagnostics**
> We agree our quantitative metrics could have been more extensive. We have therefore added a broad set of supplementary diagnostics related to isometry, including input-output Jacobian spectral statistics, layer-wise Jacobian spectral statistics, plasticity scores, effective weight ranks, weight-matrix spectral statistics, dormant neurons over time, and weight-orthogonality scores (for sample plots: https://limewire.com/d/WFKbX#wVydDx7gIl). These diagnostics are consistent with our central claim: compared to baselines, our method better preserves isometric conditioning and rank in both Jacobians and weights, maintains higher plasticity scores over time, and more strongly revives dormant neurons. This provides a more systematic account of why the method works. In terms of task performance, our method equals or surpasses prior plasticity-preserving methods.
>
> **On sufficiency**
> Our goal is to provide a *task-agnostic* perspective by characterizing plasticity in terms of intrinsic network properties. However, in our paper we argue perfect structural plasticity across all possible tasks is unattainable due to inherent null spaces in any network. This motivates our focus on (layer-wise) Jacobian isometry as a practical “best-we-can-do” target instead. We argue our work therefore provides a necessary, but not sufficient condition for plasticity (under first-order gradient-based optimization), and argue that a universally sufficient condition is unlikely to exist without restricting the task distribution. Sufficiency may become attainable under such restrictions, but this requires prior knowledge of future tasks. We hope this satisfies your remark.
>
> Additionally, we note that whenever one introduces non-linearity in the network, perfect global isometry becomes impossible. The objective is therefore to navigate the trade-off between expressivity and plasticity, and to operate near an appropriate Pareto front. We view our framework as one principled way to study this trade-off, and expect future work (possibly motivated from different angles) to be complementary rather than contradictory.
>
> **Stability and efficiency of AdamO**
> Targeting isometry/jacobian conditioning improves stability of training as it prevents pathologies such as norm blowup (related to sharp loss landscapes/large Hessian eigenvalues), reduced effective learning rate (when considering scale-invariance, cf. Lyle et al., 2024a), and poorly conditioned gradient propagation. Additionally, adamo has a low compute overhead: O(mn^2) for layer dimensions m,n (O(m^3) for square layers). This is comparable to a single forward pass, O(Bmn), when batch size B ~m. In practice, the wall-clock overhead was only a few percent.

---

> > ### Author Rebuttal · Reviewer_1wUh · 2026-04-01
> >
> > The author have partially solved my concerns and questions, however, i'd like some more evidence on Transformers empirically or theoretically, or Transformers-simplified models. I will maintain my score to positive at current time.
> >
> > Edit:  I have raised my score to Acceptance for the Transformer experiments.

---

> > > ### Author Response · Authors · 2026-04-06
> > >
> > > Thank you for your continued engagement and for encouraging us to pursue Transformer experiments. We are happy to report that we have conducted extensive experiments on Transformers on the CIFAR10 pixel-permutation benchmark. Results are available here: https://limewire.com/d/boFIc#xKmsTo42X6.
> > >
> > > **Setup.** We use a Transformer architecture of 4 attention blocks (each with 4 attention heads, followed by MLPs). We ablate across several axes central to the success of modern transformers, comparing AdamO against (1) vanilla transformer, (2) with LayerNorm, and (3) with LayerNorm + weight decay. Hyperparameter tuning was performed for all methods, and each setting is tested both with and without skip connections, as the addition of skip connections leads to different spectral dynamics. This setup provides a comprehensive comparison of AdamO in terms of stability and performance.
> > >
> > > **Results.** AdamO consistently outperforms all baselines across configurations and learning rates. Two findings stand out:
> > >
> > > 1. **AdamO outperforms LayerNorm and Weight Decay combined.**
> > > AdamO improves upon LN+WD in both stability and performance, and maintains high Jacobian effective rank and low Jacobian condition number. Moreover, at learning rates where LN(+WD) fails to learn/stabilize well (and vanilla baselines diverge), AdamO still learns successfully. These findings suggest that AdamO could substitute two separate, widely-used interventions (LN+WD) that are central in modern Transformer training (though further work is needed to verify this at larger scales). Our experiments also show that even without skip connections, Adamo outperforms skips+LN+WD. This is consistent with our central thesis: directly targeting isometry (i.e., the full singular value spectrum) improves over partial interventions.
> > >
> > > 2. **Consistency across architectural variants.**
> > > The benefits of AdamO hold both with and without skip connections. Skips+LN+WD also enable somewhat stable learning at higher learning rates (which is to be expected, as these three are used today to stabilize Transformers), but AdamO, with or without skips, still outperforms them, which is encouraging. Moreover, in the no-skip setting, where training is notoriously difficult and LN/WD alone often proves insufficient, AdamO enables stable learning, underscoring its strength as an architecture-agnostic intervention.
> > >
> > > Put briefly: in all settings, AdamO performs at least as well and is more stable than baselines, both with and without skips.
> > >
> > > We believe these results strengthen our claims and demonstrate that our framework extends effectively to attention-based architectures. We have incorporated the results into the revised manuscript alongside a brief spectral analysis of Transformer architectures (applying the work of Noci et al. (2022) and Cowsik et al. (2025) to isometric residuals/weights). We leave a larger-scale study, including on Large Language Models and fine-tuning, for future work.
> > >
> > > ---
> > >
> > > Noci, L., Anagnostidis, S., Biggio, L., Orvieto, A., Singh, S. P., & Lucchi, A. (2022). Advances in Neural Information Processing Systems, 35, 27198-27211.
> > >
> > > Cowsik, A., Nebabu, T., Qi, X., & Ganguli, S. (2025). Physical Review E, 112(5), 055301.

---

### Official Review · Reviewer_CctP · 2026-03-12

**Soundness:** 3
**Presentation:** 4
**Significance:** 3
**Originality:** 4
**Overall Recommendation:** 5
**Confidence:** 3

**Summary:**

"Loss of plasticity" is an issue in which a neural network loses capacity during training and cannot then be trained on new data; it is a widely discussed topic in the reinforcement learning literature.  This submission aims to connect this problem to a heretofore-distinct line of work on how to ensure trainability at initialization.  That line of work has argued that a sufficient means of ensuring trainability at initialization is to enforce that the layerwise Jacobians are all isometries (Jacobian has all eigenvalues equal to 1) or nearly so.  Accordingly, this paper proposes several mechanisms for enforcing or regularizing towards this condition: (i) enforcing strict orthogonality of the weight matrices (ii) regularizing towards orthogonality of the weight matrices, and (iii) the groupsort activation function, which is guaranteed to have orthogonal Jacobians everywhere.  The experiments show that these methods can fix the plasticity loss problem, and generally at least one of them works better than the baselines.

**Compliance With Llm Reviewing Policy:**

Affirmed.

**Final Justification:**

I'll raise my score to "accept."

**Key Questions For Authors:**

none

**Limitations:**

The authors didn't explicitly discuss limitations, but I didn't get the sense that the paper was overclaiming in any way.

**Strengths And Weaknesses:**

I think it's nice that the paper draws a connection between the plasticity loss literature and the dynamical isometry literature; this potentially makes for a valuable perspective on plasticity loss.  I also think it's interesting and valuable that the various proposed approaches can successfully address the plasticity loss issue, and it's also nice that the paper interprets prior plasticity loss methods in the same framework.  The main downside of the paper is that none of the proposed methods is likely the 'optimal' way to address plasticity loss, and none of the proposed methods uniformly dominated the others in experiments.  So, before this paper, we had several ways of addressing plasticity loss, and now after this paper we have several more.  The theoretical perspective is interesting and _potentially_ promising, but I would still say that we lack a precise, necessary-and-sufficient understanding of how to prevent plasticity loss.  (Of course, that goal may be very challenging, and may not be possible at the present stage of development of deep learning theory.)  Overall, to me, the paper lies somewhere between a "weak accept" and an "accept."

---

> ### Author Rebuttal · Authors · 2026-03-31
>
> Thank you for the thoughtful and constructive review. We are glad you found our paper valuable, and that the main contributions, and in particular the connection between plasticity loss and dynamical isometry, are clear and appreciated.
>
> **On the absence of a single “optimal” method**
> We agree that our proposed methods often perform equally. This may make it less obvious which approach to actually use. However, we emphasize that the role of strict Newton–Schulz (NS) orthogonalization in our work is at this point primarily *diagnostic*: it serves mainly to validate the theoretical link between plasticity and (approximate) isometry under controlled conditions. In its current form, strict NS orthogonalization is computationally expensive (O(m^3) for layer size m, but several passes are required). Compared to recent approximate NS-based methods (e.g., in Muon with Polar Express), we apply more iterations to enforce tighter orthogonality, and additionally use both forward-pass NS and a periodic backward pass (to condition the “raw” weights as well). This combination increases wall-clock time by roughly 1.5-2x, although we believe this could be heavily improved (this could be something for future work).
>
> For practical use, we recommend AdamO, which provides a favorable balance between simplicity and performance. It requires no extra lines of code to set up (basically your regular optimizer init: `optimizer = adamo(lr=3e-4, lambda=1e-2)`), and we plan to upstream implementations to JAX/Torch to further reduce adoption friction. Moreover, adamo has low compute overhead: (~O(m^3) for layer width m). This is comparable to a single forward pass, O(Bm^2), when batch size B ~m. In practice, the wall-clock overhead was only a few percent.
>
> A similar trade-off appears on the activation side. While GroupSort achieves near-ideal properties (e.g., high effective rank and stable gradient propagation), it slightly underperforms ReLU in our current setting. We attribute this to two factors:
>   (1) ReLU offers higher expressive efficiency per parameter;
>   (2) adamo/NS successfully revives dormant neurons.
>
> We therefore currently recommend adamo+ReLU for practical use. We believe that the above discussion will be useful for other readers too (and will include it in the paper), and that this point was not clear yet in our work. Thank you for highlighting it, it improves the paper.
>
> **A necessary-and-sufficient characterization of plasticity**
> Our goal is to provide a *task-agnostic* perspective by characterizing plasticity in terms of intrinsic network properties. However, in our paper we argue perfect structural plasticity across all possible tasks is unattainable due to inherent null spaces in any network. This motivates our focus on (layer-wise) Jacobian isometry as a practical “best-we-can-do” target instead. We argue our work therefore provides a necessary, but not sufficient condition for plasticity (under first-order gradient-based optimization), and argue that a universally sufficient condition is unlikely to exist without restricting the task distribution. Sufficiency may become attainable under such restrictions, but this requires prior knowledge of future tasks.
>
> Finally, we note that whenever one introduces nonlinearities (even including GroupSort), perfect *global* isometry becomes impossible. The objective is therefore to navigate the trade-off between expressivity and plasticity, and to operate near an appropriate Pareto front. We view our framework as one principled way to study this trade-off, and expect future work (possibly motivated from different angles) to be complementary rather than contradictory.

---

> > ### Author Rebuttal · Reviewer_CctP · 2026-04-04
> >
> > Thanks for the follow-up discussion.  I agree that the paper would be more interesting if you could frame it as arguing in favor of one particular recipe rather than as exploring several strategies.

---

> > > ### Author Response · Authors · 2026-04-06
> > >
> > > Thank you once more for the constructive engagement; the current version has thus been updated to reflect our discussion.
> > >
> > > We also want to highlight that our paper is more than the specific plasticity remedy alone. It provides a unifying geometric characterization of plasticity (prior work typically studies isolated failure modes), enabling more principled and architecture-agnostic interventions on the root causes of plasticity loss. Central to the theory was our extensive “necessity/sufficiency” argument (now rephrased as such in the revision). Adamo flows directly from this theory, and additionally provides a novel mechanism for reviving dormant neurons (a critical problem in deep RL in its own right). Furthermore, our work provides a deeper interpretation of prior plasticity methods, and identifies that each targets only a restricted aspect of isometry, and thus is only partially successful, which is demonstrated in our experiments, strengthening our framework as a mature contribution to the field.
> > >
> > >
> > > We hope this broader perspective is useful context as you finalize your assessment, and we remain happy to discuss any remaining questions. We also note that results on Transformers have now been added, showing improved stabilization and performance over baselines (cf. our response to reviewer 1wUh).

---

### Official Review · Reviewer_z1gy · 2026-03-12

**Soundness:** 3
**Presentation:** 3
**Significance:** 2
**Originality:** 2
**Overall Recommendation:** 3
**Confidence:** 3

**Summary:**

This paper proposes a isotrmetry regulariaztion for preventing loss of plasticity, an interesting problem

**Compliance With Llm Reviewing Policy:**

Affirmed.

**Final Justification:**

See acknowledgement: I still feel that lop is an oversolved problem, but I thank the authors for the detailed rebuttal. I have increased my score to 3. In my opinion, I think this result is borderline-ish, and I would really want to give it a score of 3.5 if that is possible. But in any case, I do not mind accepting this paper

**Key Questions For Authors:**

See weakness

**Limitations:**

Well discussed

**Strengths And Weaknesses:**

Strength: The idea of linking istrometry to loss of plasticity is interesting

Weakness: I think this result is not quite novel, and the contribution is quite incremental. Conceptually, geometry-based studies of loss of plascity exist and should have been discussed in this paper: arxiv.org/abs/2309.16932, arxiv.org/abs/2408.15495

Given so many methods that have been proposed to solve it, the comparison with other methods is quite minimal in this paper, and the performance gain is quite unclear. If the authors aim to propose a method, it should be evaluated in far more systematic and convincing ways. But his also points to another novelty problem, LoP has been solved so many times, and the progress this paper makes seems fundamental

Also, the figures should have bigger fontsizes

---

> ### Author Rebuttal · Authors · 2026-03-31
>
> We thank the reviewer for their evaluation and for recognizing the potential interest of linking isometry to loss of plasticity. We respectfully disagree, however, with the assessment that the paper is incremental.
>
> - Our core contribution is not a standalone loss of plasticity remedy (which we believe would still have its merits), but a unifying geometric framework: we propose a formal characterization of plasticity under gradient descent, grounded in the geometry of the network map. This is missing in prior work, which typically defines plasticity informally (e.g., as “trainability” or “the capacity to meaningfully adapt to new experiences”) or studies specific failure modes in isolation. Guided by our theoretical analysis, we design a practical regularization framework with a novel decoupled optimizer, and evaluate it in both supervised and reinforcement learning settings (also unveiling a successful mechanism for reviving dormant neurons, a critical problem in the field of reinforcement learning on its own). In addition, Section 5 reinterprets several influential plasticity-preserving methods (on which our own methods improve) through the unifying lens of dynamical isometry, and shows each targets only a restricted aspect of isometry rather than full-spectrum control, which our theory identifies and our method addresses. We argue these contributions are not incremental, and no existing work already provides them.
>
> - We also respectfully disagree that the paper does not position against prior plasticity methods. Section 5 is dedicated precisely to this: NaP, ReDo, L2-Init, SN reg, WD, LN are analyzed as preserving only partial notions of isometry, whereas our framework highlights the importance of full-spectrum control. We will make this distinction clearer in the revision.
> We moreover provide an empirical comparison with other baselines in our supervised and RL experiments.
>
> - The cited works (Ziyin, 2024; Ziyin et al., 2025) on symmetry absorbing states under SGD/weight decay are interesting and we will add them to the discussion, but we believe them to be significantly different to our work and not make our contribution redundant. The author studies symmetry absorbing states and operates mainly under the assumption of weight decay, which they argue is a main driver of symmetry absorption. They mention potential loss of plasticity downstream in the paper as a result of these absorptions. In contrast, our work does not assume weight decay. It instead presents a structural account of plasticity and applies generally. Moreover, Ziyin suggests a symmetry-breaking approach through fixed small random perturbations on the weights. We argue this may avoid strict 0 singular values but would still yield poorly conditioned Jacobians (small singular values, low effective rank), and therefore will still lead to slow/reduced learning. Our characterization of plasticity captures this explicitly, and our methods target these pathologies fully and directly; not only in the minimal sense of Ziyin. We also want to note that Ziyin et al. do not compare their symmetry breaking algorithm against other plasticity preserving methods, and their own results actually show that weight decay partially preserves plasticity (which was known in the literature, and to which we add, cf Ch.5). This indicates that symmetry absorption is not the full picture for plasticity, but rather a single limiting case. However, we believe the discussion to be relevant and will clarify the relationship in our work.
>
> - You mention that our methods should “be evaluated in far more systematic and convincing ways.” To address this concern, we have added a broad set of supplementary diagnostics, including input-output Jacobian spectral statistics, layer-wise Jacobian spectral statistics, plasticity scores, effective weight ranks, weight-matrix spectral statistics, dormant neurons over time, and weight-orthogonality scores (for sample plots: https://limewire.com/d/WFKbX#wVydDx7gIl). These diagnostics are consistent with our central claim: compared to baselines, our method better preserves conditioning and rank in both Jacobians and weights, maintains higher plasticity scores over time, and more strongly revives dormant neurons. This provides a more systematic account of why the method works. In terms of task performance, our method equals or surpasses prior plasticity-preserving methods.
>
> - We respectfully disagree that loss of plasticity is a solved problem. It is a rather recent and still very lively field. The reviewer’s own cited papers are recent examples of continuing work in this area. Our contribution is not another isolated fix, but a unifying account that explains several existing methods and improves upon them.
>
> - We agree with the comment on figure readability and have increased the font sizes.

---

> > ### Author Rebuttal · Reviewer_z1gy · 2026-04-01
> >
> > I still feel that lop is an oversolved problem, but I thank the authors for the detailed rebuttal. I have increased my score to 3. In my opinion, I think this result is borderline-ish, and I would really want to give it a score of 3.5 if that is possible. But in any case, I do not mind accepting this paper

---

> > > ### Author Response · Authors · 2026-04-06
> > >
> > > We are glad that you indicate all your (technical) concerns are adequately addressed, and we appreciate the updated score to 3.5. The remaining concern seems to be whether, as you state, plasticity loss is an “oversolved problem”. We want to highlight that reviewer qbZH is explicitly stating it is “an active and open area in 2026”, and that the other reviewers are also highlighting the novelty of connecting dynamical isometry and plasticity, leading to new insights and algorithms.
> > >
> > > More broadly, plasticity remains one of the central (alongside catastrophic forgetting) open research directions in continual learning, with regular contributions at top venues. Our paper offers a unifying geometric framework that reinterprets and improves upon prior methods (i.e., not an isolated fix or architecture-specific). We believe this is precisely the kind of contribution the field needs as it matures.
> > >
> > > We hope this perspective provides useful additional context, and in any case we thank the reviewer for their engagement.

---

### Official Review · Reviewer_qbZH · 2026-03-12

**Soundness:** 3
**Presentation:** 4
**Significance:** 4
**Originality:** 4
**Overall Recommendation:** 5
**Confidence:** 4

**Summary:**

This paper proposes that plasticity loss in the continual learning setting can be understood through the lens of dynamical isometry. They argue that plasticity loss is driven by layerwise Jacobian singular values drifting away from 1 during training. This leads to a cascade: when layer weights drift away from isometry, the layer Jacobian singular values spread out (the input-output Jacobian becomes ill-conditioned), the parameter Jacobian rows become close to dependent, the NTK smallest eigenvalue collapses, so gradient descent can't correct errors in some directions and this prevents the network from learning new tasks.

They formalize plasticity in terms of the NTK: they argue that the smallest eigenvalue of the NTK is what controls worst-case learnability (how well can gradient descent move in the direction that is most difficult to move in). They also show that isometry does not imply a loss of expressivity by showing that isometric almost-everywhere function classes can be universal approximators.

The paper proposes AdamO, which adds a Gram-deviation regularization penalty which encourages each weight matrix's singular values towards 1. AdamO decouples the regularization term similar to how AdamW decouples the weight decay term from Adam, which prevents the adaptivity and momentum from interfering with the isometric regularization term. This also provides a mechanism for reviving dead ReLU units, since the penalty produces a nonzero rotational gradient even for neurons with zero task gradient. Whereas the original dynamical isometry literature focused on initialization choices and activation function design in order to achieve dynamical isometry at initialization and then hope it is preserved throughout training, this paper actively enforces it throughout continual learning via regularization.

They include supervised continual learning experiments (split CIFAR-100 and permuted CIFAR-100) and show AdamO outperforms Adam, AdamW, ReDo and Shrink & Perturb, along with RL experiments (MinAtar) using PPO + AdamO showing that AdamO continues learning across thousands of tasks while baselines stall.

They track the NTK and layer singular value distributions throughout training showing that AdamO prevents collapse.

**Compliance With Llm Reviewing Policy:**

Affirmed.

**Key Questions For Authors:**

1. What is the computational overhead of the AdamO regularization term (both computation and memory)?

**Limitations:**

Yes

**Strengths And Weaknesses:**

Strengths:
- Overall this is a very nice paper and I recommend acceptance. It makes nice connections between principled theoretical work on dynamical isometry (which was most active in ~2017-2020) with the plasticity loss problem in continual learning (which is an active and open area in 2026).
- It has a clean theoretical narrative building a clear causal chain from Jacobian conditioning to NTK collapse to plasticity loss. This framing is novel (as far as I know) and well-motivated.
- The theory motivates a principled method with a simple implementation: AdamO is essentially Adam with a decoupled regularization term motivated by the dynamical isometry framing. The property that the regularization term can also revive dead ReLU neurons is more satisfying than heuristic fixes to recycle dead neurons.
- Good breadth of experiment tasks (supervised CL and RL) along with nice mechanistic validation (tracking the NTK eigenvalues and singular value distributions to show that AdamO is actually preventing collapse).
- The related work section is especially nice by providing a reinterpretation of related work on plasticity loss through the lens of dynamical isometry, rather than simply cataloguing related methods.

Weaknesses:
- The experiments use MLPs and CNNs and don't include any results on Transformers. The dynamical isometry literature originated on these simple architectures, but it's less clear how the principles apply in Transformers or how the empirical results would extend to modern architectures. For example, there may be some ways that Transformers affect the optimization geometry or normalization layers help control the weight matrix conditioning, which might be relevant.
- CIFAR-100 and MinAtar are moderate in scale. It would be nice to see larger experiments if feasible.
- It seems plausible that there is a tension between maintaining plasticity and remembering what you learned, so an analysis of whether the isometry regularization term increases catastrophic forgetting would be interesting.
- It would be good to include an ablation over the $\lambda$ hyperparameter that controls the isometry regularization penalty strength, to show whether it requires careful tuning on each task / domain.
- Minor: The paper doesn't discuss the computational overhead of AdamO, which requires computing the Gram-devation penalty for every layer at every step.
- Minor: The isometry almost-everywhere universal approximation proof uses GroupSort activations, but the experiments use ReLU, so the theoretical guarantee that isometry doesn't hurt expressivity doesn't directly apply to their experimental setting. This gap could be acknowledged better in the paper.

---

> ### Author Rebuttal · Authors · 2026-03-31
>
> Thank you for your thoughtful and encouraging review. We sincerely appreciate your positive assessment and your careful engagement with the paper.
>
> **On Transformers**
> Our paper emphasizes the importance of both well-conditioned weights and activations. At the same time, attention is often a (highly) contractive, and thus non-isometric, operation. This is a key reason larger transformers rely on skip connections, which improve gradient propagation outside the residual branches (though they in turn require normalization for stability because the singular values of the skip and residual paths interact). Given these differences, we considered a full isometry-based analysis of transformers beyond the scope of this work and listed it as future work in the Discussion. That said, we are attempting to set up small-scale transformer experiments inspired by Osband’s recent work on Delightful Policy Gradient to study orthogonal regularization and its interaction with attention. We plan to still include these as a preliminary analysis in the paper, while leaving a thorough large-scale study to future work.
>
> **On scale**
> In addition to the supervised and MinAtar results, the appendix includes experiments on Octax, a new JAX-native benchmark suite in the spirit of ALE, comprising of a series of retro games with rich dynamics (https://github.com/riiswa/octax). We chose Octax over ALE because our continual setting requires repeated runs across all games, amounting to roughly 120M steps for a single seed of a single algorithm (and 1.2B steps for MinAtar). Without substantial GPU resources, these numbers make ALE prohibitively expensive. Octax offers similarly rich dynamics while being far more accessible computationally due to its JAX-native design, and we believe it will become a popular benchmark for RL research. For additional context, in the supervised setting, a single run for one algorithm on one benchmark with 8 seeds required on the order of tens of hours (some 50+) on an Nvidia L40S (~ comparable to an A100).
>
> **Plasticity vs. forgetting**
> Although our focus is plasticity, our method can be compatible with approaches that target forgetting, such as extra auxiliary losses (that being said, there is an unavoidable trade-off between plasticity and retention under limited capacity that no algorithm will solve). To illustrate the mild forgetting properties of our methods, consider the sequence task 1 → task 2 → task 1. On task 2, our method would likely converge to a solution still reasonably close in parameter space to the original task 1 solution, unlike (soft) reset-based methods such as ReDO or L2 init, which can discard prior information and may force task 1 to be relearned from scratch. NaP does not rely on resets, but it gradually renders certain weight directions inaccessible, making return to the original task 1 solution increasingly difficult. By contrast, our Jacobian conditioning generally permits backtracking to earlier solutions (barring adversarial cases). We will add a small experiment on this in the appendix to evaluate this.
>
> **Regularization strength**
> We agree that an ablation is valuable and will include one. In practice, we found the regularizer to be quite robust to reasonable tuning ranges.
>
> **Computational overhead**
> The regularizer has computational cost O(mn^2) for layer dimensions m,n (O(m^3) for square layers). This is comparable to a forward pass, O(Bmn), when batch size B ~m. In practice, the wall-clock overhead was only a few percent. The memory overhead is one additional Gram matrix per layer, corresponding in practice to roughly 4-5% for some typical experiments. We will include a quantitative table. Thank you for highlighting this point.
>
> **ReLU/GroupSort expressivity.**
> Standard ReLU networks are universal function approximators and therefore expressive. However, as we note, consistent with Pennington et al. (2018), they cannot be generally isometric because ReLU is contractive. As a result, we can only directly control weight isometry rather than full network isometry. Nevertheless, we identify the discussed ReLU revival mechanism that compensates for part of this loss, which helps explain why the method performs so well in practice.

---

> > ### Author Rebuttal · Reviewer_qbZH · 2026-04-03
> >
> > Thanks to the authors for these responses. The intuition about why the analysis on Transformers is challenging is helpful. The additional experiments and tables you plan would be good for the paper. I maintain my score of 5.

---

> > > ### Author Response · Authors · 2026-04-06
> > >
> > > Dear reviewer, results on Transformers have now been added. Results are available here: https://limewire.com/d/boFIc#xKmsTo42X6. AdamO shows improved stabilization and performance over baselines, and early results suggest it could serve as a replacement for the modern LN+WD combo. For a more complete discussion, we refer to our reply to reviewer 1wUh. We thank you again for your engagement and your positive assessment.

---

### Decision · Program_Chairs · 2026-04-30

**Decision:**

Accept (regular)

**Comment:**

There is general consensus among the reviewers that this paper makes an interesting connection between dynamical isometry and plasticity in learning. The paper reads well and the experiments (while the scope can be improved) demonstrate interesting phenomena. The main criticism from the negative review rests upon novelty and comparison with prior related literature. I believe that the rebuttal has addressed this and if incorporated in the revision will make the paper better positioned.